# What Matters More For In-Context Learning under Matched Compute Budgets: Pretraining on Natural Text or Incorporating Targeted Synthetic Examples?

## Abstract

Does explicitly exercising the induction circuit during pretraining improve in-context learning (ICL), or is natural text sufficient, when compute is held constant (iso-FLOPs)? To test whether targeted synthetic data can accelerate the emergence of induction heads and enhance ICL performance, we introduce *Bi-Induct*, a lightweight curriculum that injects forward-copy (*Induction*), backward-copy (*Anti*, as a control), or a balanced mix, into the pretraining stream. We conduct iso-FLOPs pretraining across models from 0.13B to 1B parameters, evaluating effects across three axes: (i) few-shot performance on ICL benchmarks, (ii) head-level telemetry, and (iii) held-out language modeling perplexity. Our findings challenge the intuition that early induction circuit activation directly translates to better ICL. While Bi-Induct accelerates induction head emergence at smaller scales, this does not consistently yield better few-shot generalization. On standard LM benchmarks, Bi-Induct matches natural-only training; on function-style ICL probes, the 1B natural-only model performs best. Stress tests (e.g., label permutation, HITS@1 vs. HITS@3, 1 vs. 10 shots) preserve these trends. Telemetry reveals that larger models trained only on natural text develop broader and earlier-peaking induction heads, despite seeing no explicit induction patterns. Anti-induction data fails to elicit meaningful activation. Perplexity penalties from synthetic data shrink with scale, suggesting that larger models can absorb non-natural patterns with minimal cost. Crucially, ablating the top 2% of induction heads per layer degrades ICL more than random ablations, especially for natural-only models, indicating more centralized, load-bearing circuits. Bi-Induct variants exhibit more redundant induction activity, pointing to different circuit utilization patterns. Overall, we find that inducing activation is not sufficient: improvements in ICL hinge on whether these circuits become functionally necessary. These results underscore the importance of mechanism-aware pretraining diagnostics and data mixtures that foster *load-bearing*, not merely present, structure.

## 1 Introduction

Transformer language models learn a simple copy pattern early in training: when a token $A$ reappears in context, the model increases the probability of the token that followed the previous $A$. Prior work identified a two-head motif implementing this behavior and linked it to in-context learning on pattern-matching tasks (Olsson et al., 2022). Despite its simplicity, this motif typically emerges only after many billions of tokens, well after the first training-loss plateau. Theoretical and empirical studies frame the delay as a phase transition (Chen et al., 2024; Edelman et al., 2024). Accelerating the onset of this transition could reduce compute and expose internal circuits for analysis earlier.

A practical way to approach this is to intervene on the *data* rather than the *architecture* or *objective*. In contrast to synthetic-task-only plateau-shortening studies (Kim et al., 2025) and objective-level interventions such as multi-token prediction (Gloeckle et al., 2024), we adopt a data-rewrite perspective that is easy to deploy at scale: inject a small fraction of targeted inputs into the pretraining stream that selectively excite the putative induction mechanism while keeping the architecture and objective fixed. Concretely, we replace a small slice of natural tokens with synthetic copy snippets that cleanly exercise the copier-selector behavior of induction (forward copy) and anti-induction (backward copy). Copy-style cues are the canonical probe for the induction circuit and are widely

used to measure it (Olsson et al., 2022; Nanda & Bloom, 2022). While other synthetic families have been studied (for example n-gram statistics (Edelman et al., 2024), p-hop tasks (Sanford et al., 2024), and intrinsic tasks (Gu et al., 2023)), copy snippets align most directly with the hypothesized mechanism and with distributional properties such as burstiness that correlate with the rise of in-context learning (Chan et al., 2022).

These considerations lead to a single, testable question: *Under iso-FLOPs, what is more effective for in-context learning—pure natural text, or natural text with a small fraction replaced by directional copy snippets that directly exercise the induction circuit?*

To answer this, we introduce *Bi-Induct*, a lightweight curriculum that interleaves short duplicate-span snippets with natural text during early training. We evaluate three variants under matched compute: forward induction, backward anti-induction, and a balanced mix where the direction is chosen *independently at each injection*. We assess the impact of our pretraining strategies on three axes: (i) downstream ICL performance on standard few-shot benchmarks; (ii) a mechanistic metric, namely the concentration of the top 2% most active attention heads per layer; and (iii) language modeling quality via held-out perplexity. We conduct experiments at three model scales (0.13B, 0.5B, 1B) under iso-FLOPs constraints, also using the 0.13B model as a design space to explore curriculum parameters, including data mix ratios, and span lengths effects.

**Method and scope:** We implement *Bi-Induct* as a lightweight curriculum, introducing a linearly annealed mixture of synthetic directional-copy data during early pretraining which we compare against a pure-natural baseline under iso-FLOPs conditions. To disentangle model capability from calibration, we evaluate few-shot ICL performance alongside two diagnostics: held-out perplexity (as a quality guardrail) and head-level telemetry measuring attention to copy patterns. Our objective is not to optimize benchmark scores, but to understand when and how targeted data augmentation induces load-bearing behavior for ICL.

**Contributions:**

- **Matched-compute comparison across scales:** We conduct an iso-FLOPs comparison of pure-natural pretraining vs. a small directional-copy replacement, run at 0.13B, 0.5B, and 1B parameters on the same corpus with identical token budgets and training steps per scale. We report per-task and composite few-shot ICL results alongside a perplexity guardrail and attention-head telemetry (forward/anti-copy scores) at the final checkpoint.
- **Key empirical finding: Early induction $\neq$ better ICL.** While *Bi-Induct* accelerates and broadens induction-head activity at 0.13B and 0.5B parameters, this does not translate into superior ICL endpoints; at 1B, the natural-only baseline outperforms *Bi-Induct* on function-style probes.
- **Induction-head ablations revealing circuit load-bearing:** Ablating the top 2% induction heads per layer reduces ICL more than matched random ablations at all scales. The *largest relative drop is seen in the natural-only baseline*, while *Bi-Induct* variants degrade less, suggesting a more redundant or distributed use of the induction circuit under synthetic augmentation.
- **Design-lab ablations (0.13B):** We use the 0.13B model as a design lab to sweep curriculum parameters, including copy-span length, initial synthetic-to-natural ratio, and annealing schedules, to identify scaling-relevant settings.
- **ICL robustness diagnostics:** We evaluate model robustness through shot-count variation, label-permutation stress tests, and decision-rule sensitivity via HITS@$k$ on function-style probes.

For a concise glossary of terms and symbols, see Appendix A.

## 2   RELATED WORK

**Induction heads and the mechanics of ICL:** The induction-head motif—a two-head circuit that matches a repeated cue and copies its following token—was introduced by Olsson et al. (2022), who provided multiple converging tests linking it to the rise of in-context learning (ICL). Follow-up theory and controlled synthetic-task studies characterize the behavior as a phase transition: on Markov-chain data, models move from uniform predictions to unigram heuristics and then abruptly to bigram induction (Edelman et al., 2024). Provable analyses show that even shallow transformers implement generalized induction via a copier-selector-classifier pipeline, tightening the link between optimization dynamics and the circuit (Chen et al., 2024). At scale, targeted head ablations support causality:

removing a small fraction of high-score induction heads reduces few-shot gains by up to ∼32% on abstract pattern tasks and weakens benefits on NLP tasks (Crosbie & Shutova, 2025). Open suites and tools (e.g., Pythia and TransformerLens) have made these effects reproducible across model sizes (Biderman et al., 2023; Nanda & Bloom, 2022).

**Anti-induction and copy-suppression circuits:** Beyond forward copying, models host heads that suppress copying or implement the backward, "anti-induction" direction. Work on negative heads explains copy suppression as a coherent mechanism that interacts with induction patterns (McDougall et al., 2023). Large-scale empirical reports find a pretrained asymmetry—transformers are stronger at forward induction than backward copy—an imbalance that targeted fine-tuning can reduce while isolating distinct head families (Veitsman et al., 2025). In parallel, Wang et al. (2025) mechanistically link the 'repetition curse' to over-dominant induction heads and propose head-level regularization to restore output diversity.

**Curricula that accelerate circuit emergence:** A growing line of work seeks to shorten the ICL plateau. Training on diverse ICL tasks in parallel reduces plateau length and eases optimization relative to single-task settings (Kim et al., 2025). Orthogonal to data choice, multi-token prediction modifies the objective to encourage longer-range patterns and shows favorable development of induction-like behaviors together with efficiency gains (Gloeckle et al., 2024). However, these studies concentrate on *forward* induction and are typically evaluated in *synthetic-task-only* training regimes rather than natural-language pretraining, or they rely on objective/architectural changes rather than data-only interventions in end-to-end runs. To our knowledge, they also do not systematically induce or measure *anti*-induction emergence.

**Embedding induction heads in downstream systems:** Architectural and application work has begun to 'install' $n$-gram induction heads to stabilize in-context RL and reduce data needs, demonstrating practical leverage of the circuit in agents (Zisman et al., 2025).

**Data rewriting:** Beyond filtering, recent work rewrites pretraining text to shift style and structure before learning. Rephrasing the Web (WRAP) uses instruction-tuned models to paraphrase web pages into target styles, yielding ∼3× faster pretraining on noisy corpora, lower perplexity, and modest zero-shot gains under matched compute budgets (Maini et al., 2024). Nguyen et al. (2025) pursue a related transform-and-retain strategy focused on discarded low-quality documents. Fujii et al. (2025) expand the rewrite paradigm beyond style, reporting boosts on math and code. In parallel, large open datasets such as RefinedWeb show that aggressive deduplication and domain organization improve pretraining without synthetic rewrites, positioning data rewriting as complementary to quality and mixture knobs (Penedo et al., 2023).

**Circuit discovery and emergence shaping:** Mechanistic interpretability maps internal circuits via activation interventions (Zhang & Nanda, 2024), probing (Gurnee et al., 2023), and increasingly, sparse-autoencoder features (Cunningham et al., 2023). Our focus is earlier in the lifecycle: shaping the data distribution so that desirable circuits appear sooner and more predictably, then verifying the link with head-level telemetry.

***Summary:*** Existing work investigates *circuit discovery/emergence shaping* and, separately, *data rewriting*, but rarely bridges the two—i.e., using mechanistic insight to design pretraining data and evaluating the outcome under matched-compute conditions. We make that link explicit: we target a canonical ICL circuit (forward and backward induction) with the minimal inputs that excite it (directional copy snippets), and compare *pure natural pretraining* to *Bi-Induct*, a small, linearly annealed replacement policy, under iso-FLOPs conditions on the same corpus. We measure effects behaviorally (few-shot ICL benchmarks) and mechanistically (top 2% head concentration by layer), alongside a perplexity guardrail. Unlike prior curricula that emphasize forward copy alone, we study a symmetric forward/backward curriculum side by side to ask whether targeted copy signals are more valuable than additional natural tokens at the same compute.

## 3 BI-INDUCT

We investigate the effect of *data rewrites* on circuit emergence: we *interleave* synthetic copy snippets into the pretraining stream to explicitly teach the copy algorithm. Bi-Induct has two primary variants that differ only in the direction of the copy cue (forward vs. backward). A third variant, *balanced*, flips a coin between forward and backward injections to provide a mixed signal.

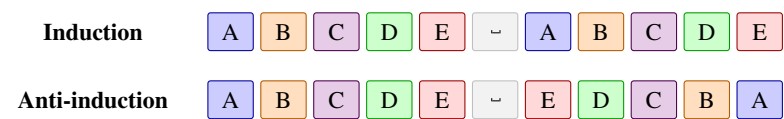

Figure 1: Examples of copy-style snippets injected into the pretraining stream. Each snippet is a span of $L$ random non-special tokens, followed by a separator, then either the same span (induction) or the reversed span (anti-induction). Colors align repeated tokens across the two halves. The illustration uses $L=5$ for clarity.

## 3.1 SYNTHETIC SNIPPET CONSTRUCTION

Let $\mathcal{V}$ be the tokenizer vocabulary and let $\mathrm{BPE}(\cdot)$ be the tokenizer. For a span length $L$, we first sample a token span

$$S = (s_1, \ldots, s_L), \qquad s_i \sim \mathrm{Uniform}(\{\lfloor 0.05|\mathcal{V}|\rfloor, \ldots, \lfloor 0.95|\mathcal{V}|\rfloor\}),$$

which avoids special/rare IDs. We use a single space as a neutral separator, $\mathrm{SEP} = \mathrm{BPE}(\texttt{" "})$.

**Forward/induction (Figure 1, top):**

$$\mathrm{Induction}(S) = [S \,\|\, \mathrm{SEP} \,\|\, S].$$

**Backward/anti-induction (Figure 1, bottom):**

$$\mathrm{Anti}(S) = [S \,\|\, \mathrm{SEP} \,\|\, \mathrm{reverse}(S)].$$

**Balanced/mix of forward and backward injections:** On each injection, flip a fair coin to choose between forward or backward.

Each snippet has length $\ell_{\mathrm{snip}} = 2L + |\mathrm{SEP}|$ (e.g., $2L+1$ when SEP is a single space).

## 3.2 CURRICULUM SCHEDULE AND INJECTION RULE

We interleave snippets on the fly during streaming pretraining. Let $m_0$ be the initial mix ratio[1] and $T_a$ an anneal budget (in natural tokens). After $t$ natural tokens have been seen, the instantaneous injection probability is

$$m(t) = \max\{\, m_0 \cdot (1 - t/T_a),\, 0 \,\}.$$

On each natural example, draw $u \sim \mathrm{Uniform}(0, 1)$. If $u < m(t)$, we first yield one synthetic snippet (depending on the current $\texttt{synthetic\_task} \in \{\texttt{induction}, \texttt{anti}, \texttt{balanced}\}$), then yield the natural tokenized sequence. *Else* ($u \geq m(t)$), we emit only the natural tokenized sequence. This implements a light interleave rather than full replacement and keeps the natural distribution dominant.

**Expected injection budget ($\bar{m}$):** With a linear anneal $m(t) = m_0(1 - t/T_a)$ for $t \in [0, T_a]$ and $m(t)=0$ afterwards, the *average* injection rate over the anneal is $m_0/2$. Let $T_{\mathrm{base}}$ be the natural-token budget of the run. The fraction of injected snippets over the *whole* run is therefore:

$$\bar{m} \;=\; \frac{1}{T_{\mathrm{base}}} \int_0^{T_{\mathrm{base}}} m(t)\, dt \;=\; \begin{cases} m_0/2, & T_a \geq T_{\mathrm{base}}, \\ m_0 \dfrac{T_a}{2 T_{\mathrm{base}}}, & T_a < T_{\mathrm{base}}. \end{cases}$$

**Why this schedule?** (i) *Front-loading the signal:* Induction circuits typically emerge after the first loss plateau; concentrating copy cues early helps trigger the phase transition without interfering with late-stage calibration; (ii) *Stability:* A linear anneal avoids abrupt distribution shifts and exposes a single knob ($m_0$) for clean sweeps; and (iii) *Compute considerations:* Under standard packing, snippets can share sequences with natural text so the incremental token cost scales with $\ell_{\mathrm{snip}}$ rather than a full segment. We *enforce iso-FLOPs across conditions* (fixed sequence length and optimizer steps), so any potential savings from aggressive packing are intentionally not exploited.[2]

---

[1] Or mix ratio, for short.

[2] We fix compute to avoid conflating efficiency optimizations with capability changes; our focus is whether targeted directional copy improves ICL *at the same compute*.

Table 1: Model presets used in experiments. Attention uses head dimension 64; #heads = hidden/64 and #KV heads = $\max(1, \lfloor \#\text{heads}/4 \rfloor)$.

| Model | Layers | Hidden | MLP (intermediate) | Head dim | #Attn heads | #KV heads |
|-------|--------|--------|--------------------|----------|-------------|-----------|
| 0.13B | 12 | 768 | 3,072 | 64 | 12 | 3 |
| 0.5B | 30 | 1,024 | 4,096 | 64 | 16 | 4 |
| 1B | 30 | 1,536 | 6,144 | 64 | 24 | 6 |

## 4 EXPERIMENTAL SETUP

**Model:** We use a causal decoder-only Transformer with rotary position embeddings (ROPE, $\theta$=10,000), pre-norm residual blocks, and a gated MLP with SiLU activation (SwiGLU). Self-attention uses *grouped key-value attention* (GQA): for head dimension 64, we set number of attention heads = hidden/64 and number of KV heads = $\max(1, \lfloor \#\text{heads}/4 \rfloor)$. We train in **bfloat16** with context length 1,024 and *untied* input/output embeddings.[3] Hidden sizes, heads, KV heads, layer counts, and proportional MLP widths are shown in Table 1.

**Pretraining data:** We pretrain on the deduplicated THE PILE dataset (Gao et al., 2020) in streaming/shuffled mode. A stable MD5-based hash assigns a fixed held-out evaluation slice so train/eval partitions remain identical across runs; we set this slice to **0.2%** of the corpus which corresponds to roughly **0.4B** tokens. Tokenization truncates to 1,024 tokens per sequence. Synthetic snippets are interleaved by the Bi-Induct iterator as described in Section 3.

**Training recipe:** We train all model presets with peak learning rate 1e-3 with linear warmup of 3% of the token budget then cosine decay for the rest. We optimize using AdamW ($\beta_1 = 0.9$, $\beta_2 = 0.999$, weight-decay 0.1), with each update consuming $2^{16}$ tokens. Following the Chinchilla compute-optimal rule (Hoffmann et al., 2022), we set the total token budget to $T_{base} \approx 20N$ tokens, where $N$ is the number of model non-embedding parameters. We compute the baseline update count as $U = \lceil T_{base}/2^{16} \rceil$ and keep $U$ identical across all *Bi-Induct* curricula at a given scale to enforce iso-FLOPs. We monitor training loss and evaluate perplexity at the final checkpoint on a held-out split of the natural corpus (without synthetic snippets).

**Variants:** We compare four variants, namely BASELINE (no snippets), INDUCTION (forward copy), ANTI (backward copy), and BALANCED (coin flip per injection).

**Metrics and guardrails:** We assess Bi-Induct along three complementary axes: (i) *downstream ICL performance* on standard few-shot benchmarks; (ii) *mechanistic telemetry* that targets the intended circuit (induction and anti-induction heads); and (iii) *quality* guardrails. Benchmarks are run few-shot (5-shot by default); we also include function-style tasks from the Todd et al. (2024) suite at 10-shot, evaluated with HITS@1 accuracy to stress simple copy and selection behaviors. For mechanism evidence, we compute per-head copy scores and, at the final checkpoint, report the **top 2%** of heads per layer (and their concentration) for both induction and anti-induction, contrasting Bi-Induct curricula with the baseline. As a guardrail, we report held-out language modeling perplexity (PPL). Table 2 summarizes each metric and its preferred direction; for detailed definitions see Appendix B.

**Design lab at 0.13B:** We use the **0.13B** model as a design lab to select the operating point for larger-scale runs. Unless noted otherwise, all ablations use a **2.6B** token budget, a **1024** context length, and a **linear anneal over the full budget**.

- **Span length** ($L$): We sweep $L \in \{5, 20, 500\}$ under Bi-Induct and find that $L$=20 offers the best trade-off between few-shot ICL performance and held-out perplexity. For detailed analysis see Section C.1, Appendix C.
- **Mix ratio** ($m_0$): With span fixed at $L$=20, we sweep the initial mix ratio $m_0 \in \{25\%, 50\%, 100\%\}$ (linearly annealed to zero over the full budget). We select **50%** because it yields stronger and more concentrated induction-head activity (top-2% concentration by layer) while maintaining competitive ICL and PPL; see also Section C.2, Appendix C.

---

[3]Our architecture largely follows the Mistral-7B design (decoder-only, pre-norm, RoPE, SwiGLU, GQA) (Jiang et al., 2023).

Table 2: Summary of outcome metrics and guardrails. Full definitions in Appendix B.

| Family | Metric | What it measures / protocol | Better |
|---|---|---|---|
| Standard LM benchmarks | ICL composite (macro) | Unweighted mean across 3-shot tasks (MMLU, ARC-C, BoolQ, LAMBADA, PIQA; plus others where used). Accuracy or exact match per task; averages over demo seeds. | ↑ |
| | Per-task scores | Per-benchmark few-shot evaluation (3-shot by default). Report mean over seeds with the benchmark's standard metric (Acc or EM). | ↑ |
| Function probes | ICL composite (macro) | Unweighted mean across ICL tasks probing string manipulation/selection (capitalize_*, next_item, word_length, alphabetically_*, choose_*). Default 10-shot; metric is HITS@1 accuracy; seeds averaged. | ↑ |
| | Per-task scores | Per-probe 10-shot (unless stated) with HITS@1 accuracy; seeds averaged. | ↑ |
| Mechanistic telemetry | Head copy score (top 2% per layer) | Per-head induction and anti-induction copy scores at the final checkpoint; report, for each layer, the top 2% heads and their concentration to reveal circuit strength and specialization vs. baseline. | ↑ |
| Quality | Perplexity (held-out) | PPL on a fixed 0.2% THE PILE validation slice (stable hash), same tokenizer and context across runs; mean over seeds at iso-FLOPs. | ↓ |

**Summary and choice for scaling:** For the main experiments across **0.13B**, **0.5B**, and **1B**, we adopt $L=20$ and $m_0=50\%$, which we linearly anneal to 0 over each model's full training token budget (anneal horizon $T_a = T_{\text{base}}$)[4].

## 5 NATURAL-ONLY VS. DIRECTIONAL COPY-SNIPPET MIX (ISO-FLOPS)

### 5.1 DOWNSTREAM ICL PERFORMANCE

We evaluate two groups of tasks under iso-FLOPs and average over three seeds: (i) *standard LM benchmarks* (14 tasks; e.g., MMLU, BBH, GSM8K, ARC-C, HellaSwag) and (ii) *function-style probes* from the Todd et al. (2024) suite (19 tasks). For the full inventory of *both* groups see Table 5, Appendix B. We report macro-averages per group here and provide per-task scores in Table 8, Appendix D.1.

**Standard LM benchmarks:** Figure 2a reports 5-shot macro-averages across 14 benchmarks. At each scale, at least one *Bi-Induct* variant matches or very slightly exceeds the natural-only baseline: at 0.13B, *Anti* is closest (22.5 vs. 22.7); at 0.5B, *Induction* leads (23.9 vs. 23.6); at 1B, *Balanced* is on par or marginally higher (24.3 vs. 24.2). Within measurement noise, copy-snippet curricula are *largely performance-neutral* at these scales (i.e., neither clearly degrading nor reliably improving downstream ICL).

**Function probes (Todd et al., 2024):** Figure 2b shows 10-shot macro-averages over 19 probes. At 0.13B and 0.5B, *Bi-Induct* variants are comparable to baseline; at 1B, the natural-only baseline is clearly stronger across the suite.

Robustness checks (1-shot evaluation, label permutation, and shifting the decision rule from HITS@1 to HITS@3) shift absolute scores, but preserve the cross-regime ordering; for details see Appendix D.2.

### 5.2 MECHANISTIC TELEMETRY

Figure 3 visualizes layerwise copy scores for the top 2% attention heads per layer (with a floor of one head per layer to avoid sampling artifacts). Three clear patterns emerge.

*(i) Emergence timing:* At 0.13B and 0.5B, Bi-Induct variants show earlier induction-head emergence than the baseline (by roughly 3 and 2 layers, respectively). At 1B, the trend reverses: the baseline is the first to form clear induction peaks (around layers 10-11) and its early peaks are higher than

---

[4]Annealing over the full budget avoids introducing a second scheduling timescale, reduces re-tuning, and keeps the recipe portable across scales.

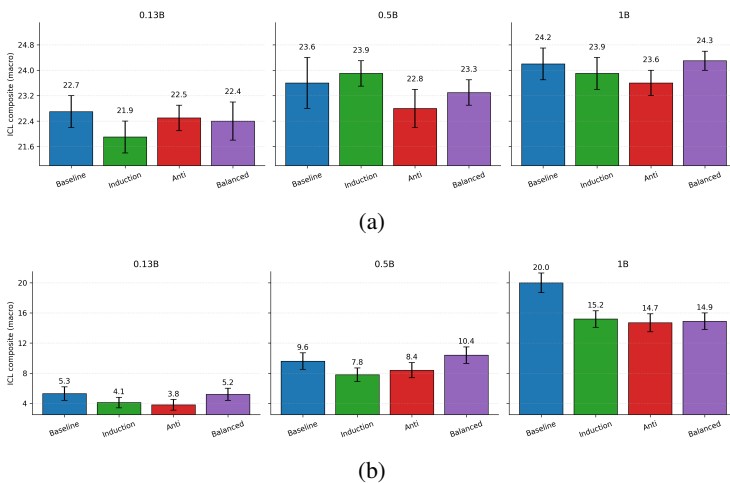

Figure 2: ICL Composite (macro) across two evaluation families: (a) Standard LM benchmarks; (b) Todd et al. (2024)'s function-probe suite. Each panel groups by model size (0.13B, 0.5B, 1B), bar colors by training regime (Baseline, Induction, Anti, Balanced); error bars show $\pm 1$ s.d. For per-task results see Appendix D.1, Table 8.

any Bi-Induct variant. For anti-induction, absolute scores are small at all scales; the largest peak we observe is $\approx 0.04$ at 0.5B (Induction curriculum), followed by $\approx 0.02$ at 1B (Baseline). In keeping with Veitsman et al. (2025), forward-induction heads dominate in pretrained LMs; in our runs, even the *Anti* curriculum did not materially increase anti-induction copy scores. *Notably, the strongest induction activity is concentrated in mid layers, in keeping with prior observations of where induction heads typically emerge* (Olsson et al., 2022).

*(ii) Peak strength:* The maximum normalized induction score reaches values close to 1.0 at 0.13B and 0.5B, but stays well below 0.5 at 1B. Thus, even when induction emerges early at 1B (Baseline), its strongest heads are less polarized than at smaller scales.

*(iii) Spread:* We count heads with a positive copy score among those selected by our per-layer top-2% criterion (Section B.2). Using the best-performing *Bi-Induct* variant at each scale for a like-for-like comparison (Balanced at 0.13B/0.5B; Induction at 1B) we observe: for 0.13B, Baseline 3 vs. Balanced 5; for 0.5B, Baseline 7 vs. Balanced 8; for 1B, Baseline 12 vs. Induction 6. In short, *Bi-Induct* tends to yield earlier and slightly broader induction activity at 0.13B/0.5B, whereas at 1B the natural-only Baseline shows the broader spread.

**Link to ICL performance:** These mechanistics results line up with the aggregate ICL results. On standard LM benchmarks (Figure 2a), macro ICL composites are similar across curricula at each scale: when Bi-Induct makes induction emerge earlier/stronger (0.13B/0.5B), performance is comparable to Baseline; when Baseline emerges earlier (1B), Bi-Induct is still comparable. A plausible explanation is a *path shift*: larger models can route more prediction mass through FFN/residual pathways, so head-peak magnitude and spread are not the sole determinants of few-shot accuracy on these tasks—especially because many of these benchmarks are knowledge and calibration-heavy, where FFN 'key-value' memories are known to contribute substantially (Geva et al., 2021). In contrast, on Todd et al. (2024)'s function-style suite (Figure 2b), the 1B Baseline's earlier and broader induction spread coincides with a clear performance advantage over all Bi-Induct variants, suggesting these probes are more sensitive to the presence/strength of explicit copy heads.

Across scales, ablating the top-2% induction heads per layer at evaluation decreases the ICL composite more than ablating an equal number of random heads (Table 3). The effect is largest for the natural-only Baseline, and remains visible, though smaller, for all Bi-induct variants, a pattern that is consistent with either redundancy or a more distributed implementation of the behavior (random head ablations sometimes yield small improvements, as expected from noise/regularization effects). Detailed per-task comparisons for clean runs vs. induction-head and random-head ablations can be found in Table 12, Appendix E.

Table 3: Percent change in the ICL composite on the function-probe suite of Todd et al. (2024) when ablating either the top-2% highest-scoring induction heads per layer($\Delta_{\text{induct}}$) or an equal number of random heads ($\Delta_{\text{rand}}$), each measured relative to the model's clean run. Negative values indicate a drop in accuracy; positive values indicate an improvement.

| Model | Baseline | | Induction | | Anti-induction | | Balanced | |
|---|---|---|---|---|---|---|---|---|
| | $\Delta_{\text{induct}}$ | $\Delta_{\text{rand}}$ | $\Delta_{\text{induct}}$ | $\Delta_{\text{rand}}$ | $\Delta_{\text{induct}}$ | $\Delta_{\text{rand}}$ | $\Delta_{\text{induct}}$ | $\Delta_{\text{rand}}$ |
| 0.13B | $-22.6\%$ | $+17.0\%$ | $-4.9\%$ | $+7.3\%$ | $-5.3\%$ | $+5.3\%$ | $-19.2\%$ | $0.0\%$ |
| 0.5B | $-14.6\%$ | $-4.2\%$ | $-10.3\%$ | $+3.8\%$ | $-8.3\%$ | $-1.2\%$ | $-12.5\%$ | $0.0\%$ |
| 1B | $-19.5\%$ | $-4.0\%$ | $-14.5\%$ | $-2.6\%$ | $-12.9\%$ | $+0.7\%$ | $-8.7\%$ | $-4.0\%$ |

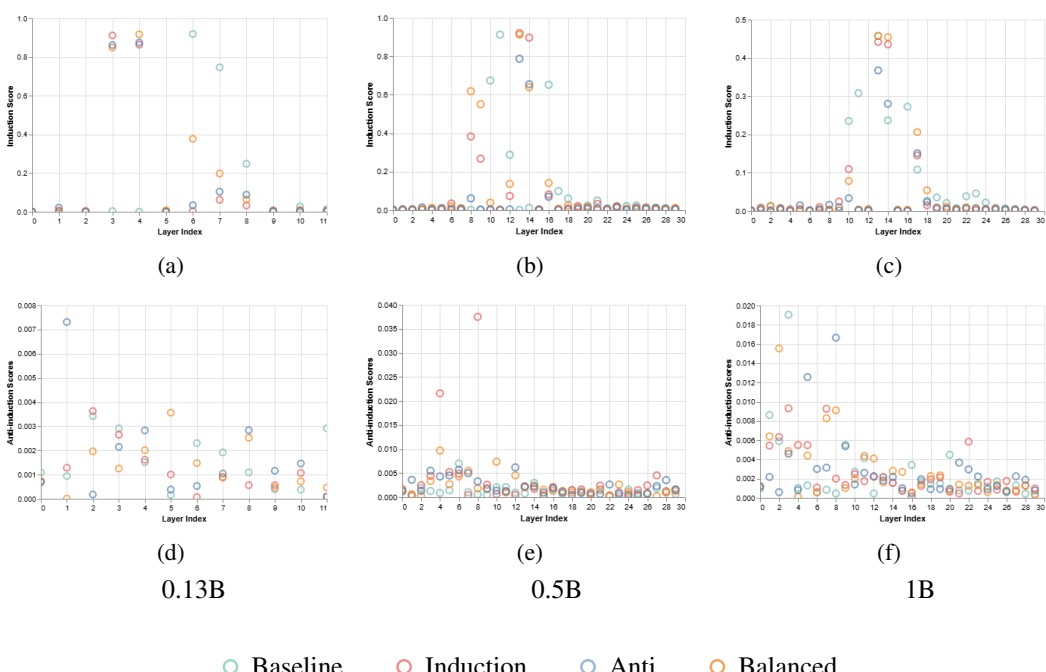

Figure 3: Layer-wise copy-head telemetry. Top row: induction scores; bottom row: anti-induction scores. For each layer we plot the best-scoring head (top 2% by score with a floor of one head per layer), averaged over three seeds, for the 0.13B, 0.5B, and 1B models. Head counts for each model are given in Table 1.

**What might drive the 1B-scale behavior?** Two non-exclusive factors may contribute, both consistent with prior literature: (1) *Width-dilution*: the 1B model has the same depth as the 0.5B model but a larger hidden size and more attention heads (24) per layer. As a result, copy behavior may be spread across more heads, reducing the peak score of any single head even if the behavior is present.[5] (2) *Pathway shift*: larger models may increasingly leverage FFN and residual pathways, reducing reliance on localized, high-scoring induction heads (Geva et al., 2021).

Overall, the mechanistic readout suggests that *Bi-Induct consistently accelerates and broadens induction activity at smaller scales*, but *at 1B the natural-only baseline exhibits earlier and broader consolidation of induction*, aligning with its stronger performance on the Todd et al. (2024) suite. In contrast, standard LM few-shot appear largely insensitive to these differences, likely due to the availability of alternative computation pathways.

---

[5]Prior work finds substantial head redundancy and small subsets of specialized/important heads (Voita et al., 2019; Michel et al., 2019; Olah et al., 2020). In wider models, this redundancy may spread copy behavior across more heads, lowering any single head's score (a speculative 'Width-dilution' effect).

Table 4: Held-out perplexity (PPL ↓) on the fixed THE PILE eval split at iso-FLOPs. Values are averaged over three seeds. For each model size, curricula use a **50%** mix ratio linearly annealed over the full training budget.

| Curriculum | 0.13B | 0.5B | 1B |
|---|---|---|---|
| Induction | *25.8* | *17.9* | *14.9* |
| Anti-induction | *26.2* | *18.2* | *14.9* |
| Balanced | *26.2* | *18.2* | *14.9* |
| Baseline | 21.8 | 16.0 | 14.1 |

## 5.3 GUARDRAIL: LANGUAGE MODELING PERPLEXITY

Table 4 reports held-out perplexity (mean over three seeds) under iso-FLOPs for each model size. We observe a consistent pattern: the *perplexity gap* between copy-snippet curricula and the baseline *shrinks with scale*, suggesting that larger models can absorb a small synthetic perturbation of the training stream without lasting calibration cost. Qualitatively, this trend is consistent with benign overfitting/double-descent intuitions: larger models can accommodate mild training perturbations while continuing to improve test loss (Nakkiran et al., 2019).

These observations validate our choice of a light, annealed Bi-Induct schedule: it preserves LM usability (competitive perplexity) and remains increasingly acceptable with scale, as the PPL gap consistently narrows across scale.

## 6 CONCLUSION AND FUTURE WORK

We posed a single matched-compute question: At iso-FLOPs, does pure natural-text pretraining outperform a curriculum that explicitly targets the induction circuit via synthetic copy snippets? Using Bi-Induct (with forward, backward, or balanced directional copy injections), we evaluated models from 0.13B to 1B parameters, combining few-shot ICL performance with head-level telemetry and held-out perplexity as diagnostic tools.

**Key finding: Natural data yields load-bearing heads, Bi-Induct induces more distributed, redundant heads.** Bi-Induct consistently accelerates and broadens induction-head activity. In contrast, the natural-only baseline relies on a smaller subset of highly active heads. As can be seen from Figure 3 and Table 3, ablating top induction heads leads to larger performance drops in the natural-only model (evidence that its induction heads are more load-bearing), whereas Bi-Induct spreads responsibility across more redundant 'backup' heads.

**Implications and Future Directions:** Acceleration is not the whole story: what matters is whether the mechanism becomes *load-bearing* for task success. Our results align with previous reports that induction heads can evolve into 'function heads' that are causally necessary for ICL performance (Yin & Steinhardt, 2025). This helps explain why the baseline—despite slower emergence in most of our runs, and a lack of redundancy in all—can achieve stronger endpoints: its induction circuitry is more load-bearing.

The above points to a shift in focus: from inducing activity earlier to ensuring that such activity is necessary for the task. To support this, rather than relying solely on across-scale comparisons, future work should prioritize within-scale trajectory analysis to identify when behaviors become load-bearing, crucial for connecting data rewrites to mechanistic insights. While one might speculate that a stronger Bi-Induct schedule could push induction behavior towards more load-bearing circuits, that was not our design goal. We intentionally used a light, annealed curriculum to preserve LM usability (competitive perplexity) while probing ICL endpoints. Substantially heavier synthetic schedules could plausibly succeed in forcing functional induction, but at the cost of degraded language modeling quality and less plausible training data mixes. Accordingly, our claims are scoped to realistic, usability-oriented pretraining where synthetic data is a small perturbation annealed early. We would argue that future work should be similarly scoped.

## USE OF LARGE LANGUAGE MODELS (LLMS)

We used general-purpose large language models as assistive tools for *writing* and *typesetting*. Concretely: (i) LLMs helped draft and polish prose across multiple sections (e.g., Introduction, Related Work, and Conclusion), including line-level rewrites for clarity, grammar, and flow; and (ii) LLMs assisted with LaTeX boilerplate and table scaffolding (e.g., column definitions, `\resizebox`, and `booktabs` structure) but did not determine the content of any table.

LLMs **were not** used to design experiments, analyze data, run code, generate results, or make scientific claims. All technical decisions, datasets, models, and analyses originated from the authors. Every LLM suggestion was reviewed, edited, and verified by the authors; all references and factual statements were cross-checked against primary sources.

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

## A  GLOSSARY AND TERMINOLOGY

This section defines the terms and metrics used throughout the paper. We group entries by theme for quick reference.

### COPY-STYLE CIRCUITS AND INTERPRETABILITY

**Mechanistic interpretability** The study of internal circuits and features that give rise to behavior in neural networks. Typical tools include ablations/masking, activation patching, causal tracing, and sparse autoencoders.

**Interpretability challenges** Practical difficulties include superposition (features sharing parameters), circuit non-uniqueness (multiple decompositions fit the data), intervention fragility (ablations can misattribute causality), scale transfer (circuits shift across sizes), and dataset confounds (spurious correlations masquerading as mechanisms).

**Induction head / induction circuit** A two-head attention motif that implements forward copy: when a cue token reappears in the context, attention retrieves what followed the *previous* occurrence and predicts it again. Empirically linked to few-shot pattern matching.

**Anti-induction** The mirror of induction: backward copy. Given a repeated cue, the model predicts the *preceding* token from an earlier occurrence (useful for reversal-style tasks and some code transforms).

**Copy-suppression (negative) heads** Attention heads whose contribution reduces copying (e.g., down-weights repeated spans), often interacting with induction heads to prevent degenerate repetition.

CURRICULUM AND DATA-REWRITE TERMS

**Data rewrite** Deliberate modification of a small fraction of pretraining tokens to teach a target algorithm (here, copy patterns) without changing the model architecture.

**Bi-Induct** Our symmetric copy-style curriculum that injects synthetic snippets during pretraining in one of two directions: `induction` (forward copy) or `anti` (backward copy). Injection probability linearly anneals to zero.

**Span length** ($L$) Number of random tokens in the snippet's base span before duplication or reversal (e.g., $L \in \{5, 20, 100\}$).

**(Initial) Mix ratio** Initial probability of injecting a synthetic snippet before annealing (e.g., $25\%$).

**Anneal tokens** The number of natural tokens over which the injection probability decays linearly to zero (e.g., the full 2.5B-token budget).

**"Balanced" variant** A coin-flip per injection between forward and backward copy. Used as an additional control in some ablations.

COMPUTE AND EFFICIENCY

**Chinchilla (compute-optimal) budget** The token-parameter trade-off that minimizes validation loss at fixed compute for dense decoder-only LMs. Rule of thumb: a tokens-to-parameters ratio of $\approx$ 20:1, i.e., $T \approx 20N$ (tokens $T$, parameters $N$).

EVALUATION ENDPOINTS

**ICL benchmarks (few-shot endpoints)** Standard few-shot ($k \geq 1$) tasks evaluated at the *final* checkpoint (e.g., 5-shot MMLU, ARC-C, BoolQ, LAMBADA, PIQA). We aggregate with a macro-average as the **ICL composite**. These are the *main* outcome metrics. *Regarding the standard deviation (s.d.) of the ICL composite:* In Tables 6 and 7, we compute the s.d. of the per-seed composite across seeds, which is the appropriate uncertainty. Elsewhere, for brevity, we approximate the composite's uncertainty by averaging per-task s.d.s computed across seeds; this is a readable proxy but not a pooled s.d. and it ignores cross-task covariance.

**Cross-entropy and perplexity** Language-model loss on a held-out split of the pretraining corpus. Perplexity $PPL = \exp(CE)$. Used as a quality and calibration guardrail.

# B   METRICS AND GUARDRAILS: DETAILED DEFINITIONS

## B.1   BENCHMARKS AND PROTOCOLS

**Aggregation:**   We report a *macro* ICL composite (unweighted mean across selected tasks) and per-task scores. All figures and tables show mean over seeds. Full list of benchmarks used is in table 5

**Prompting controls:**   For few-shot tasks, we fix a template and average across multiple demonstration seeds. For robustness, we randomize demonstration order and, in §D.2, evaluate sensitivity to number of shots and a label-permutation stress test.

## B.2 MECHANISTIC TELEMETRY

**Targeted circuits:** We measure two equality-based copy circuits—*induction* and *anti-induction*—highlighted in prior work(e.g., (Olsson et al., 2022; Veitsman et al., 2025)).[6] In a left-to-right causal decoder, *attention flows from the later span back to the earlier span.* Consider a repeated sequence $x = s_0\, s_1\, \ldots\, s_{L-1}\, \langle\text{sep}\rangle\, s'_0\, s'_1\, \ldots\, s'_{L-1}$ with $s'_i = s_i$:

- *Induction (forward copy).* At position $s'_i$ in the second span, the head locates the earlier repeat and retrieves payload that helps predict the *next* token $s'_{i+1}$. We operationalize this with a *next-token* alignment (defined below).

- *Anti-induction (backward copy).* At position $s'_i$, the head again locates the earlier repeat but retrieves payload that helps predict the token immediately to the *left*, $s'_{i-1}$. We operationalize this with a *same-token* alignment (defined below).

**Probe sequences:** We evaluate on 50,000 fresh copy probes disjoint from training, each built as $x = s\, \langle\text{sep}\rangle\, s$ with a uniformly sampled token span $s$ of length $L=500$[7].

**Per-head scores (how we compute them):** Let $A^{(\ell,h)} \in \mathbb{R}^{T \times T}$ be the attention map (rows = target positions, columns = source positions) for layer $\ell$, head $h$ on $x$. Index $t_i$ as the row of $s'_i$ (second span) and $m_i$ as the column of $s_i$ (first span).

*Induction (next-token) score:* Using $\mathcal{D}_{\text{next}} = \{(t_i, m_{i+1})\}_{i=0}^{L-2}$ (later $s'_i$ to earlier $s_{i+1}$),

$$\text{Score}_I(\ell, h) \;=\; \mathbb{E}_x\!\left[\frac{1}{L-1} \sum_{(t_i, m_{i+1}) \in \mathcal{D}_{\text{next}}} A^{(\ell,h)}_{t_i, m_{i+1}}\right].$$

*Anti-induction (same-token) score:* Using the same-token diagonal $\mathcal{D}_{\text{same}} = \{(t_i, m_i)\}_{i=0}^{L-1}$ (later $s'_i$ to earlier $s_i$),

$$\text{Score}_A(\ell, h) \;=\; \mathbb{E}_x\!\left[\frac{1}{L} \sum_{(t_i, m_i) \in \mathcal{D}_{\text{same}}} A^{(\ell,h)}_{t_i, m_i}\right].$$

**Higher is better** for both $\text{Score}_I$ and $\text{Score}_A$ (stronger, more localized copy behavior).

**Top 2% concentration by layer:** Let $H_\ell$ be the heads in layer $\ell$ and $k_\ell = \max\{1, \lceil 0.02\,|H_\ell|\rceil\}$. For $Score \in \{Score_I, Score_A\}$, let $\text{Top}_\ell(S)$ be the $k_\ell$ heads with largest $S(\ell, h)$. We report the mass share

$$\text{MassShare}^{(Score)}_\ell = \frac{\sum_{h \in \text{Top}_\ell(Score)} Score(\ell, h)}{\sum_{h \in H_\ell} Score(\ell, h)},$$

and the layer mean $\overline{Score}_\ell = \frac{1}{|H_\ell|} \sum_{h \in H_\ell} Score(\ell, h)$. Larger values indicate stronger specialization (copy mass concentrated in a few heads).

## B.3 LANGUAGE MODELING QUALITY

**Perplexity:** We compute cross-entropy and PPL on a fixed 0.2% THE PILE validation slice (stable hash partition), at iso-FLOPs and identical tokenization settings across runs.

## C ABLATION STUDY

### C.1 SPAN LENGTH

We begin by testing how the snippet span $L$ affects outcomes. At **0.13B**, with a fixed initial mix of 25% linearly annealed over the full token budget, we sweep $L \in \{5, 20, 500\}$ and report two

---

[6]See also (Wang et al., 2025; Yin & Steinhardt, 2025).

[7]We evaluate with a span length of $L = 500$ (rather than $L = 20$) to reduce potential confounds from the *Bi-Induct* pretraining curriculum, which used $L = 20$.

endpoints of practical interest—(i) a 5-shot ICL composite over five standard LM benchmarks and (ii) held-out LM perplexity (PPL). We defer the function-probe suite of Todd et al. (2024) to the cross-scale experiments, where relative differences are more interpretable and the added compute is justified; the span-length results for this subsection are summarized in Table 6.

Across curricula, $L=20$ is a stable operating point that balances ICL and calibration: for *Induction*, 31.9 ICL / 23.9 PPL (vs. 30.7/23.8 at $L=5$ and 31.8/24.0 at $L=500$); for *Anti*, 32.1/24.0 (vs. 31.6/23.8 at $L=5$, 31.7/24.5 at $L=500$) respectively; for *Balanced*, 31.2/24.0 (vs.31.4/23.8 at $L=5$, 32.0/24.0 at $L=500$). Very short spans ($L=5$) underperform on the ICL composite, while very long spans ($L=500$) offer no consistent ICL gain and tend to slightly worsen PPL. Hence we adopt $L=20$ for the remaining experiments. Beyond the ICL / PPL balance, shorter spans are operationally attractive: they minimize snippet length $2L+|\text{SEP}|$, which reduces potential overhead and makes it easier to *pack* snippets alongside natural sequences to exploit variable-length kernels— yielding compute savings when such packing is enabled.[8]

## C.2 Mix Ratio

We fixed the anneal to the *full* training budget (2.6B token, following the Chinchilla parameter-token rule of thumb (Hoffmann et al., 2022)), held the span length at $L=20$, and swept the initial mix ratio over $\{25\%, 50\%, 100\%\}$. Table 7 reports full per-task results.

**Mechanistic readout:** Figure 4 summarizes layerwise copy-head activity across mix ratios. We quantify head quality in two complementary ways: (i) *spread*, the number of heads per model whose induction score is non-zero (and, for comparability, the count above a fixed "specialization" threshold $> 0.5$); and (ii) *peak sharpness*, the maximum head score in each condition. We also note *concentration* in depth (whether peaks cluster in the canonical mid-layers).

**Does synthetic injection improve induction-head quality vs. baseline?** By counts above the 0.5 threshold, yes up to moderate mixes. The baseline shows 2 specialized heads. With **Induction** snippets we observe $\{3, 3, 1\}$ specialized heads at $\{25\%, 50\%, 100\%\}$ mixes, **Balanced** yields $\{4, 4, 1\}$, and **Anti** yields $\{3, 1, 2\}$. By peak sharpness, **Balanced-25%** attains the highest induction head, with Induction and Anti close behind; at $\geq 50\%$ mixes, Balanced and Baseline retain similar peaks while Induction then Anti trail.

**Does the curriculum create anti-induction heads?** No. Even under Anti mixes, anti-induction scores remain far below the specialization threshold; the best peaks are $\approx 0.01$ (at 50% Anti and in Baseline), indicating no robust anti-induction circuit emerges.

**How does mix ratio affect spread and depth concentration (induction)?** For **Induction**, spread rises from 25% to 50% ($3{\to}5$ heads) then contracts at 100% (2). For **Balanced**, spread is largest at 25% (5), then declines (4 at 50%, 2 at 100%). For **Anti**, induction heads peak at 25-50% (4 each) and drop to 2 at 100%. Depthwise, specialized heads shift deeper as mix increases: clusters are earlier at 25% (layers $\sim$2-4), mid-depth at 50% (layers $\sim$4-6), and later at 100% (around layer $\sim$8).

**Takeaways:** Baseline naturally forms a few strong induction heads. Adding snippets increases the *number* of specialized induction heads up to moderate mixes ($\leq 50\%$); higher mixes reduce spread and push peaks deeper. None of the curricula—especially Anti—produces meaningful anti-induction heads.

# D In-Context Learning Capability

## D.1 In-Context Learning Performance

In Table 8 (summarized in Figure 2a and Figure 2b; see §5.1), we report *per-task* few-shot ICL performance across scales for two families: (i) 14 standard LM benchmarks/tasks and (ii) 19 function-style probes from the Todd et al. (2024) suite. For the standard LM benchmarks we use **3-shot** evaluation; for the Todd et al. (2024) suite we use **10-shot** evaluation. (Table 5 lists all tasks and

---

[8]*All reported results are at iso-FLOPs; we do *not* take packing credits in our comparisons. Packing is a deployment optimization, not part of the evaluation protocol.

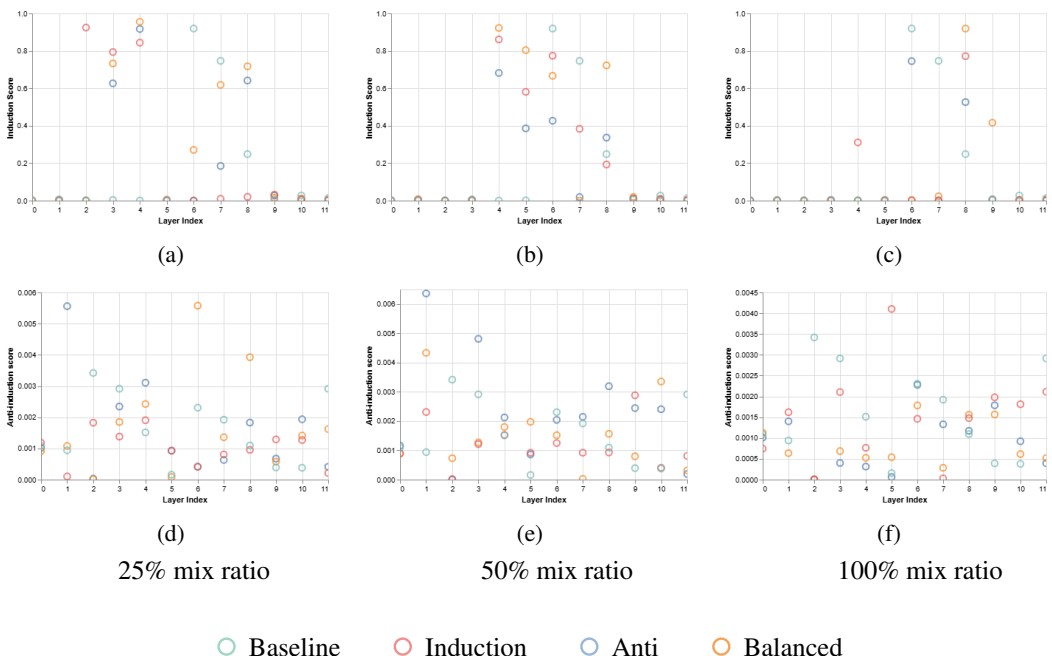

(a)                              (b)                              (c)

(d)                              (e)                              (f)

25% mix ratio              50% mix ratio              100% mix ratio

○ Baseline       ○ Induction       ○ Anti       ○ Balanced

Figure 4: Layer-wise copy-head telemetry. Top row: induction scores; bottom row: anti-induction scores. For each layer we plot the best-scoring head (top 2% by score with a floor of one head per layer), averaged over six seeds, for the 0.13B model with initial mix ratios: 25%, 50%, and 100% . Head counts for each model are given in Table 1.

provides a brief description of each.) Unless otherwise noted, metrics are accuracy (ACC) or exact match (EM) as standard, and all results are averaged over three seeds.

Because any $> 0$ shot setting exercises in-context learning, we also study **1-shot** sensitivity for the same tasks/benchmarks in Appendix D.2.

## D.2    IN-CONTEXT LEARNING ROBUSTNESS

### D.2.1    SENSITIVITY TO NUMBER OF SHOTS

We assess how the ICL results in §5.1 (with details in Appendix D.1) vary with the number of in-context demonstrations. Concretely, we change the evaluation from the main-text setting (3-shot for standard LM benchmarks and 10-shot for function-probe tasks) to a unified 1-shot setting for both families. Summaries appear in Figure 5a (standard LM) and Figure 5b (function probes); per-task scores are in Table 9.

For the standard LM benchmarks, moving to 1-shot produces negligible changes across all model sizes (0.13B, 0.5B, 1B). In contrast, the function-style probes degrade notably at 0.5B and 1B when reduced to 1-shot, while the 0.13B model shows only a small drop. This scale-dependent sensitivity aligns with prior observations that larger models more reliably use the demonstration label→token mapping (and thus benefit from more shots), whereas smaller models often gain primarily from format/structure and topical priming (Wei et al., 2023; Min et al., 2022; Zhao et al., 2021). Consistently, our label-permutation stress test shows minimal impact at 0.13B but clear degradation at 0.5B/1B, indicating that bigger models lean more on the (now corrupted) mapping signal.

Across shot conditions, the BASELINE vs. BI-INDUCT ordering remains stable: reducing shots changes the absolute level but not the ranking among curricula.

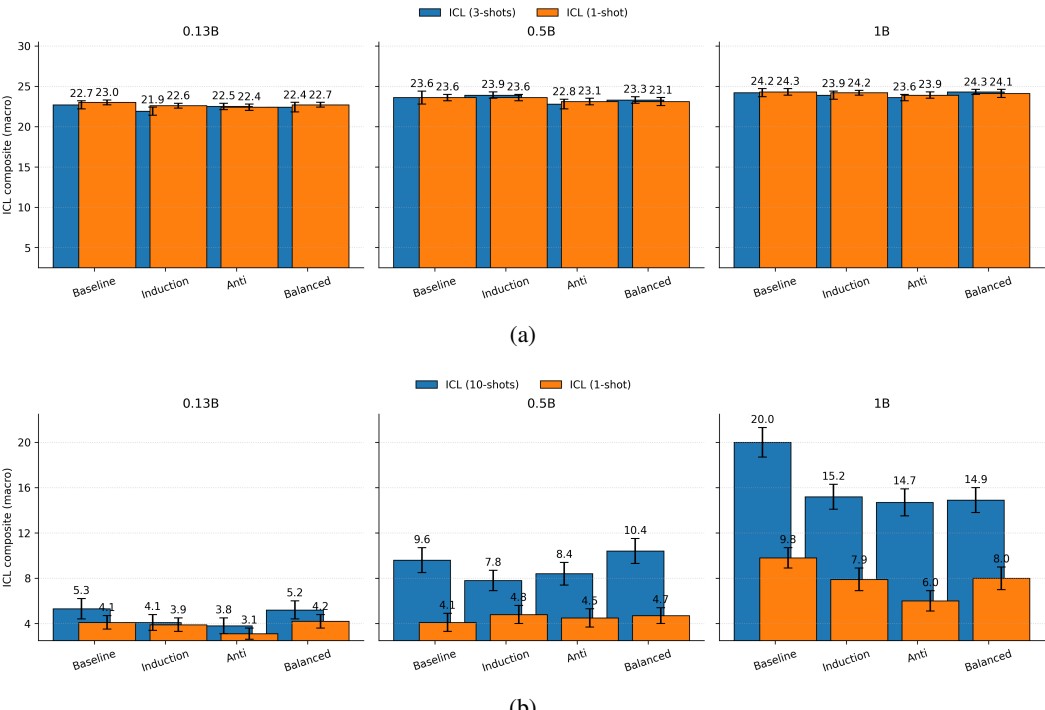

Figure 5: Sensitivity of ICL composite (macro) to the number of shots across two evaluation families: (a) standard LM benchmarks (3-shot vs. 1-shot); (b) Function-probe suite of Todd et al. (2024) (10-shot vs. 1-shot). Each panel groups models by size (0.13B, 0.5B, 1B), colors the bars by regime (Baseline, Induction, Anti, Balanced), and shows ±1 s.d. error bars.

### D.2.2 FUNCTION-PROBE TASK STRESS TESTS

Because stress tests were explicitly considered during the development of the Todd et al. (2024) function-probe suite, we evaluate two that directly target ICL robustness: (i) *label permutation* within the in-context demonstration shots, and (ii) *decision-rule sensitivity*, contrasting the commonly reported HITS@3 with our primary metric, HITS@1. These tests probe robustness to spurious label–token mappings and to the choice of evaluation rule, respectively.

**Label-Permutation Stress Test:** We stress-test in-context usage on the Todd et al. (2024) probes by randomly permuting the targets within the 10 demonstration shots (inputs unchanged) and evaluating on the true task distribution. If a model relies on the demonstrations, accuracy should drop; if it leans on parametric priors, it should be less affected. As shown in Figure 6 and Table 10, the relative ordering between Baseline and Bi-Induct variants is largely preserved. At **0.13B**, permutation produces no degradation for either, suggesting heavier reliance on parametric knowledge. At **0.5B** and **1B**, all curricula degrade, indicating increased sensitivity to the in-context mapping. Overall, Bi-Induct mirrors Baseline at each scale: robust at 0.13B and increasingly demonstration-sensitive as scale grows.

***Why permutation hurts 0.5B-1B but not 0.13B?*** At 0.13B, the robustness to label permutation suggests it benefits from demonstrations via format/topical priming and answer-frequency priors, but does not reliably exploit the label→token mapping. In contrast, 0.5B-1B models more strongly use the in-context mapping; permuting labels therefore contradicts a cue they have learned to trust, producing clear drops. This is consistent with reports that (i) labels in demos can be non-essential for smaller/less capable settings—format and priors often dominate (Min et al., 2022; Zhao et al., 2021), and (ii) the ability to override priors and follow contradictory, flipped labels emerges with scale (Wei et al., 2023).

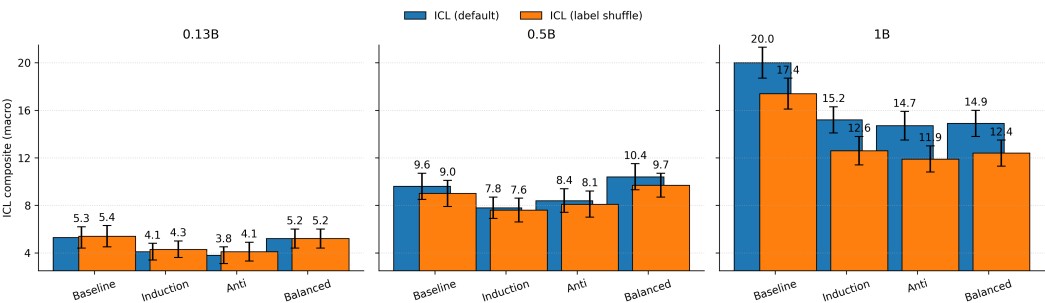

Figure 6: Function-probe suite of Todd et al. (2024) — ICL Composite (macro). Three panels (0.13B, 0.5B, 1B). For each model, bars compare *ICL (default – no label shuffle)* vs *ICL (label shuffle)* across four regimes (Baseline, Induction, Anti, Balanced). Error bars show $\pm 1$ s.d.

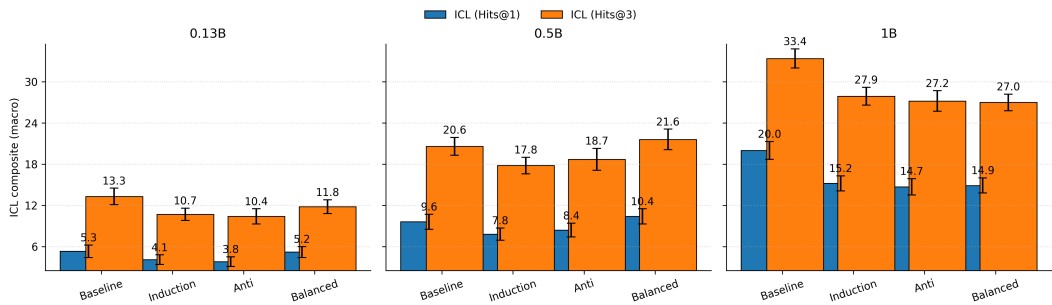

Figure 7: Function-probe suite of Todd et al. (2024) — ICL Composite (macro). Three panels (0.13B, 0.5B, 1B). For each model, bars compare *ICL (HITS@1)* vs *ICL (HITS@3)* across four regimes (Baseline, Induction, Anti, Balanced). Error bars show $\pm 1$ s.d.

**Decision-Rule Sensitivity - HITS@$k$ ($k \in \{1, 3\}$):** Following common practice for function-probe tasks in the Todd et al. (2024) suite—which reports HITS@3 (top-3 token accuracy)—we compare our primary decision rule (HITS@1) with HITS@3. ICL performance is summarized in Figure 7, and per-task HITS@3 accuracies are listed in Table 11. While HITS@3 increases absolute scores across the board, the relative ordering and gaps between variants remain effectively unchanged.

# E  INDUCTION HEAD ABLATION

We quantify how much the in-context learning (ICL) composite depends on the model's most induction-like attention heads by ablating them at evaluation time and comparing the drop to ablating an equal number of random heads.

**Selecting induction heads:** For each model, we compute a per-head *copy score* exactly as in Section B.2 and select the top 2% per layer for ablation.

**Ablation mechanism (value-stream zeroing):** At inference, for a chosen set of heads $\mathcal{S}_\ell$ in layer $\ell$, we zero their value-stream contribution before the output projection:

$$\tilde{Z}^{(\ell)} = \big[\, QKV^{(1)} \parallel \dots \parallel \underbrace{0}_{h \in \mathcal{S}_\ell} \parallel \dots \parallel QKV^{(H)} \,\big], \qquad \text{attn\_out}^{(\ell)} = \tilde{Z}^{(\ell)} W_O^{(\ell)}.$$

Queries/keys/softmax are unchanged; only the selected heads' post-attention vectors are set to zero. This follows common practice in mechanistic interpretability and avoids softmax renormalization artifacts. We compare two conditions:

- **Induct-head ablation:** zero the per-layer top-2% induction heads defined above.

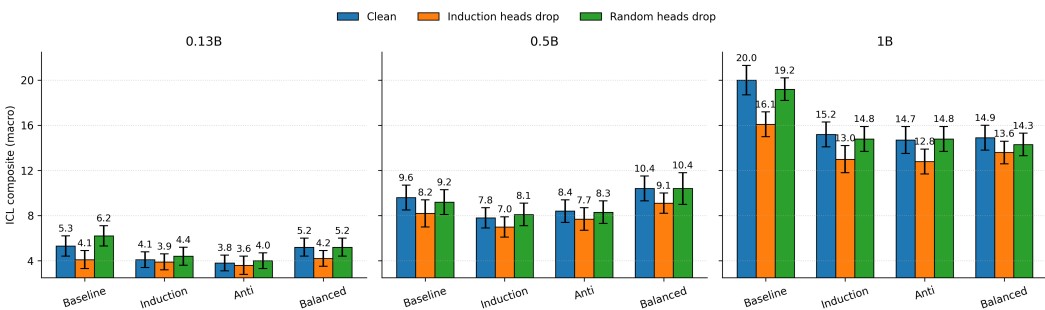

Figure 8: **Function-probe suite of Todd et al. (2024) - ICL composite under clean, induct-head ablation, and random-head ablation.** Three panels (0.13B, 0.5B, 1B), across four regimes (Baseline, Induction, Anti, Balanced). Error bars show $\pm 1$ s.d.

- **Random-head ablation:** zero the same count of uniformly random heads per layer.

We evaluate the same prompts, shots, and metrics as in the main text (Section 5.1): the macro-averaged ICL composite aggregates task scores (e.g., HITS@1 unless otherwise specified). For each model-curriculum pair we report (i) the clean score, and (ii) the percent change under each ablation relative to its own clean run:

$$\Delta_{\text{induct}} = 100 \times \frac{\text{ICL}_{\text{induct-abl}} - \text{ICL}_{\text{clean}}}{\text{ICL}_{\text{clean}}}, \qquad \Delta_{\text{rand}} = 100 \times \frac{\text{ICL}_{\text{rand-abl}} - \text{ICL}_{\text{clean}}}{\text{ICL}_{\text{clean}}}.$$

Figure 8 shows the ICL composite for clean vs. ablations across scales and curricula; per-task deltas are in Table 12.

Table 5: Benchmarks, evaluation metrics, and shot counts used to compute the ICL composite in Section 5.1.

| Benchmark / Tasks | Metric | Shots | Notes |
|---|---|---|---|
| MMLU (Hendrycks et al., 2021) | Acc | 3 | 57 subject areas; standard 5-shot setup. |
| Winogrande (Sakaguchi et al., 2019) | Acc | 3 | Commonsense coreference. |
| CommonSenseQA (Talmor et al., 2019) | Acc | 3 | Multiple choice commonsense. |
| PIQA (Bisk et al., 2020) | Acc | 3 | Physical commonsense. |
| HellaSwag (Zellers et al., 2019) | Acc | 3 | Story completion. |
| TriviaQA-Wiki (Joshi et al., 2017) | EM | 3 | Open-domain QA, Wikipedia evidence. |
| BBH (CoT) (Suzgun et al., 2022) | EM | 3 | Few hard tasks with chain-of-thought prompts. |
| OpenBookQA (Mihaylov et al., 2018) | Acc | 3 | Elementary science QA. |
| ARC-Challenge (Clark et al., 2018) | Acc | 3 | Difficult science questions. |
| GPQA (Rein et al., 2023) | Acc | 3 | Graduate-level QA. |
| GSM-8K (Cobbe et al., 2021) | EM | 3 | Math word problems with short CoT. |
| MathQA (Amini et al., 2019) | Acc | 3 | Programmatic math QA. |
| BoolQ (Clark et al., 2019) | Acc | 3 | Yes/No reading comprehension. |
| LAMBADA (OpenAI) (Paperno et al., 2016) | Acc | 3 | Cloze final-word prediction. |
| *From Todd et al. (2024) function-probe suite:* | | | |
| `capitalize` | HITS@1 Acc | 10 | Convert the entire input string to uppercase (e.g., "hello" → "HELLO"). |
| `capitalize_first_letter` | HITS@1 Acc | 10 | Uppercase the first character only; leave the rest unchanged ("alpha" → "Alpha"). |
| `capitalize_last_letter` | HITS@1 Acc | 10 | Uppercase the final character only ("gamma" → "gammA"). |
| `lowercase_first_letter` | HITS@1 Acc | 10 | Lowercase the first character only ("Alpha" → "alpha"). |
| `lowercase_last_letter` | HITS@1 Acc | 10 | Lowercase the final character only ("GammA" → "Gamma"). |
| `next_capital_letter` | HITS@1 Acc | 10 | Map an uppercase letter to its successor in the alphabet (e.g., A→B; wraparound optional). |
| `next_item` | HITS@1 Acc | 10 | Given an item from an ordered category (day, month, letter, number word), output the next item ("Monday" → "Tuesday"). |
| `prev_item` | HITS@1 Acc | 10 | As above, but return the previous item ("Tuesday" → "Monday"; wraparound for cyclic lists). |
| `word_length` | HITS@1 Acc | 10 | Return the number of characters in the input word ("token" → 5). |
| `alphabetically_first_3` | HITS@1 Acc | 10 | From a list of 3 strings, choose the alphabetically earliest. |
| `alphabetically_first_5` | HITS@1 Acc | 10 | From a list of 5 strings, choose the alphabetically earliest. |
| `alphabetically_last_3` | HITS@1 Acc | 10 | From a list of 3 strings, choose the alphabetically latest. |
| `alphabetically_last_5` | HITS@1 Acc | 10 | From a list of 5 strings, choose the alphabetically latest. |
| `choose_first_of_3` | HITS@1 Acc | 10 | From a list of 3 items, select the first item by position. |
| `choose_first_of_5` | HITS@1 Acc | 10 | From a list of 5 items, select the first item by position. |
| `choose_last_of_3` | HITS@1 Acc | 10 | From a list of 3 items, select the last item by position. |
| `choose_last_of_5` | HITS@1 Acc | 10 | From a list of 5 items, select the last item by position. |
| `choose_middle_of_3` | HITS@1 Acc | 10 | From a list of 3 items, select the middle item by position. |
| `choose_middle_of_5` | HITS@1 Acc | 10 | From a list of 5 items, select the middle item by position. |

Table 6: Span-length sweep at **0.13B** on THE PILE. All curricula are linearly annealed over the full training budget with initial mix of 25%. Results are averaged over six seeds. Evaluation is 5-shot. We report per-task accuracies, the macro ICL composite, and held-out perplexity (PPL).

| | Baseline | Induction | | | Anti-induction | | | Balanced | | |
|---|---|---|---|---|---|---|---|---|---|---|
| | - | 5 | 20 | 500 | 5 | 20 | 500 | 5 | 20 | 500 |
| MMLU ↑ | 25.2 ± 0.3 | 25.3 ± 0.5 | 25.1 ± 0.4 | 25.3 ± 0.5 | 25.3 ± 0.4 | 25.1 ± 0.5 | 24.8 ± 0.5 | 25.2 ± 0.4 | 24.8 ± 0.3 | 25.1 ± 0.2 |
| ARC-Challenge ↑ | 18.1 ± 0.6 | 18.7 ± 0.4 | 18.6 ± 1.3 | 18.1 ± 0.6 | 18.5 ± 0.7 | 18.1 ± 0.4 | 17.6 ± 0.6 | 18.6 ± 0.2 | 18.5 ± 0.6 | 17.6 ± 0.4 |
| BoolQ ↑ | 52.8 ± 4.6 | 46.2 ± 5.2 | 53.5 ± 5.9 | 53.4 ± 3.7 | 50.8 ± 6.8 | 54.5 ± 1.4 | 53.4 ± 2.9 | 51.2 ± 2.5 | 50.1 ± 3.5 | 54.7 ± 4.9 |
| LAMBADA ↑ | 7.6 ± 0.7 | 6.7 ± 0.6 | 6.4 ± 0.6 | 5.5 ± 0.2 | 6.4 ± 0.5 | 6.7 ± 0.5 | 6.2 ± 0.2 | 7.0 ± 0.1 | 6.9 ± 0.4 | 6.5 ± 0.5 |
| PIQA ↑ | 56.6 ± 0.5 | 56.5 ± 0.6 | 55.8 ± 0.5 | 55.5 ± 0.2 | 55.9 ± 0.3 | 55.9 ± 0.6 | 56.4 ± 0.8 | 56.0 ± 0.6 | 55.8 ± 0.5 | 56.0 ± 0.5 |
| ICL composite (macro) ↑ | 32.1 ± 0.9 | 30.7 ± 2.4 | 31.9 ± 1.1 | 31.8 ± 1.7 | 31.6 ± 1.2 | 32.1 ± 0.3 | 31.7 ± 1.4 | 31.4 ± 3.1 | 31.2 ± 0.7 | 32.0 ± 2.2 |
| PPL ↓ | 21.8 | 23.8 | 23.9 | 24.0 | 23.8 | 24.0 | 24.5 | 23.8 | 24.0 | 24.2 |

Table 7: Initial mix-ratio sweep at **0.13B** on THE PILE. All curricula are linearly annealed over the full training budget with span fixed at $L = 20$. Results are averaged over six seeds. Evaluation is 5-shot. We report per-task accuracies, the macro ICL composite, and held-out perplexity (PPL).

| | Baseline | Induction | | | Anti-induction | | | Balanced | | |
|---|---|---|---|---|---|---|---|---|---|---|
| | - | 25% | 50% | 100% | 25% | 50% | 100% | 25% | 50% | 100% |
| MMLU ↑ | 25.2 ± 0.26 | 25.1 ± 0.38 | 25.0 ± 0.33 | 24.9 ± 0.33 | 25.1 ± 0.48 | 25.2 ± 0.40 | 24.9 ± 0.41 | 24.8 ± 0.29 | 25.0 ± 0.27 | 24.6 ± 0.38 |
| ARC-Challenge ↑ | 18.1 ± 0.64 | 18.6 ± 1.28 | 18.1 ± 0.65 | 17.9 ± 0.89 | 18.1 ± 0.42 | 18.0 ± 0.47 | 17.5 ± 0.91 | 18.5 ± 0.58 | 18.1 ± 0.44 | 17.8 ± 0.52 |
| BoolQ ↑ | 52.8 ± 4.64 | 53.5 ± 5.92 | 49.6 ± 6.92 | 51.8 ± 4.51 | 54.5 ± 1.36 | 47.8 ± 6.78 | 51.1 ± 2.87 | 50.1 ± 3.52 | 46.9 ± 4.68 | 51.8 ± 5.37 |
| LAMBADA ↑ | 7.6 ± 0.69 | 6.4 ± 0.60 | 5.6 ± 0.50 | 4.7 ± 0.50 | 6.7 ± 0.45 | 5.6 ± 0.51 | 4.4 ± 0.45 | 6.9 ± 0.43 | 5.9 ± 0.66 | 4.6 ± 0.38 |
| PIQA ↑ | 56.6 ± 0.49 | 55.8 ± 0.48 | 55.4 ± 0.27 | 54.9 ± 0.49 | 55.9 ± 0.55 | 55.4 ± 0.37 | 55.2 ± 0.55 | 55.8 ± 0.50 | 55.8 ± 0.72 | 54.4 ± 0.45 |
| ICL composite (macro) ↑ | 32.06 ± 0.89 | 31.88 ± 1.08 | 30.76 ± 1.48 | 30.83 ± 0.98 | 32.06 ± 0.25 | 30.40 ± 1.40 | 30.62 ± 0.66 | 31.22 ± 0.69 | 30.34 ± 0.92 | 30.64 ± 1.10 |
| PPL ↓ | 21.8 | 23.9 | 26.0 | 31.4 | 24.0 | 26.4 | 32.9 | 24.0 | 26.2 | 32.8 |

Table 8: Results across model scales (0.13B, 0.5B, 1B) on THE PILE at iso-FLOPs. Copy snippets use span $L$=20. Evaluation is few-shot: 3-shot for standard LM benchmarks, and 10-shot for function-style probes. We report per-task accuracy (or EM where standard), averaged over three seeds, and the ICL composite (macro-average across tasks). Higher is better.

| | Baseline | | | Induction | | | Anti-induction | | | Balanced | | |
|---|---|---|---|---|---|---|---|---|---|---|---|---|
| | 0.13B | 0.5B | 1B | 0.13B | 0.5B | 1B | 0.13B | 0.5B | 1B | 0.13B | 0.5B | 1B |
| MMLU | 26.7 ± 0.1 | 27.5 ± 0.4 | 27.7 ± 0.0 | 25.9 ± 0.0 | 27.4 ± 0.0 | 26.6 ± 0.0 | 27.1 ± 0.0 | 26.8 ± 0.0 | 27.2 ± 0.1 | 26.2 ± 0.1 | 27.1 ± 0.2 | 27.1 ± 0.0 |
| Winogrande | 50.4 ± 0.6 | 50.8 ± 1.4 | 49.8 ± 1.1 | 47.0 ± 0.9 | 50.5 ± 1.1 | 51.1 ± 0.8 | 51.2 ± 0.9 | 50.1 ± 1.0 | 49.8 ± 1.3 | 50.9 ± 1.6 | 50.9 ± 0.4 | 51.0 ± 1.2 |
| CommonSenseQA | 20.8 ± 1.2 | 20.0 ± 1.1 | 21.2 ± 0.9 | 20.3 ± 0.3 | 20.5 ± 0.9 | 20.5 ± 0.8 | 20.8 ± 0.3 | 20.0 ± 1.2 | 20.0 ± 0.4 | 20.6 ± 0.7 | 20.5 ± 0.3 | 20.4 ± 0.4 |
| PIQA | 56.6 ± 0.3 | 58.5 ± 1.2 | 59.2 ± 0.5 | 55.0 ± 0.2 | 58.6 ± 0.1 | 58.4 ± 0.3 | 56.1 ± 0.2 | 56.9 ± 0.3 | 58.9 ± 0.6 | 55.5 ± 0.7 | 57.2 ± 0.6 | 58.4 ± 0.3 |
| HellaSwag | 26.5 ± 0.1 | 27.1 ± 0.4 | 27.8 ± 0.1 | 26.3 ± 0.1 | 26.5 ± 0.1 | 27.2 ± 0.1 | 26.4 ± 0.1 | 26.8 ± 0.1 | 27.3 ± 0.1 | 26.2 ± 0.1 | 26.5 ± 0.1 | 27.3 ± 0.1 |
| TriviaQA-Wiki | 0.1 ± 0.0 | 0.2 ± 0.0 | 0.4 ± 0.0 | 0.1 ± 0.0 | 0.1 ± 0.0 | 0.3 ± 0.0 | 0.1 ± 0.0 | 0.1 ± 0.0 | 0.3 ± 0.0 | 0.1 ± 0.0 | 0.1 ± 0.0 | 0.3 ± 0.0 |
| BBH (CoT) | 0.1 ± 0.0 | 1.5 ± 0.2 | 2.8 ± 0.0 | 0.3 ± 0.0 | 1.6 ± 0.1 | 1.2 ± 0.1 | 0.8 ± 0.1 | 0.1 ± 0.0 | 0.6 ± 0.0 | 0.1 ± 0.0 | 2.2 ± 0.0 | 4.2 ± 0.0 |
| OpenBookQA | 14.3 ± 0.6 | 14.0 ± 1.6 | 15.9 ± 1.1 | 13.9 ± 0.8 | 15.3 ± 0.1 | 15.3 ± 0.4 | 14.7 ± 0.7 | 13.9 ± 1.3 | 15.6 ± 0.4 | 13.1 ± 0.1 | 13.6 ± 1.0 | 16.5 ± 0.5 |
| ARC-Challenge | 18.6 ± 0.6 | 18.4 ± 1.1 | 18.2 ± 0.6 | 18.3 ± 0.3 | 18.5 ± 0.7 | 18.1 ± 0.4 | 17.6 ± 0.4 | 17.8 ± 0.4 | 19.0 ± 0.4 | 17.1 ± 0.8 | 17.9 ± 0.4 | 17.9 ± 0.3 |
| GPQA | 25.2 ± 2.2 | 26.1 ± 2.1 | 25.2 ± 2.0 | 23.7 ± 2.3 | 25.2 ± 1.7 | 24.3 ± 1.5 | 23.9 ± 2.0 | 24.9 ± 1.4 | 24.1 ± 1.1 | 24.1 ± 2.2 | 22.8 ± 0.8 | 24.6 ± 0.8 |
| GSM-8K | 1.5 ± 0.4 | 1.5 ± 0.3 | 1.7 ± 0.2 | 1.1 ± 0.3 | 1.5 ± 0.3 | 1.5 ± 0.2 | 1.4 ± 0.2 | 1.2 ± 0.1 | 1.6 ± 0.1 | 1.1 ± 0.4 | 1.7 ± 0.2 | 1.4 ± 0.2 |
| MathQA | 20.5 ± 0.4 | 20.9 ± 0.7 | 20.5 ± 0.7 | 19.9 ± 0.6 | 20.3 ± 0.2 | 20.7 ± 0.6 | 20.2 ± 0.3 | 20.4 ± 0.4 | 21.1 ± 0.2 | 21.0 ± 0.7 | 20.8 ± 0.4 | 21.0 ± 0.1 |
| BoolQ | 48.8 ± 0.5 | 53.4 ± 0.9 | 54.7 ± 0.2 | 49.1 ± 0.7 | 60.5 ± 0.3 | 57.0 ± 0.9 | 49.1 ± 0.8 | 52.2 ± 2.2 | 52.7 ± 0.3 | 51.4 ± 0.5 | 57.1 ± 1.0 | 58.1 ± 0.4 |
| LAMBADA (OpenAI) | 8.2 ± 0.2 | 11.0 ± 0.4 | 13.2 ± 0.1 | 5.5 ± 0.1 | 8.6 ± 0.1 | 12.2 ± 0.2 | 5.2 ± 0.2 | 8.4 ± 0.3 | 12.2 ± 0.2 | 6.0 ± 0.2 | 8.2 ± 0.2 | 12.0 ± 0.3 |
| ICL composite (macro) ↑ | 22.7 ± 0.5 | 23.6 ± 0.8 | 24.2 ± 0.5 | 21.9 ± 0.5 | 23.9 ± 0.4 | 23.9 ± 0.5 | 22.5 ± 0.4 | 22.8 ± 0.6 | 23.6 ± 0.4 | 22.4 ± 0.6 | 23.3 ± 0.4 | 24.3 ± 0.3 |
| alphabetically_first_3 | 4.9 ± 0.6 | 9.4 ± 0.6 | 19.7 ± 0.9 | 4.6 ± 0.6 | 7.3 ± 0.3 | 15.1 ± 0.9 | 3.3 ± 0.5 | 8.5 ± 0.9 | 14.4 ± 1.5 | 4.8 ± 0.6 | 11.5 ± 0.6 | 15.3 ± 1.0 |
| alphabetically_first_5 | 4.0 ± 0.6 | 6.7 ± 0.8 | 12.4 ± 1.4 | 4.2 ± 0.8 | 6.5 ± 1.0 | 10.9 ± 0.9 | 2.8 ± 0.7 | 7.4 ± 0.6 | 8.4 ± 1.1 | 3.9 ± 1.0 | 8.2 ± 0.8 | 10.2 ± 0.7 |
| alphabetically_last_3 | 3.4 ± 0.5 | 10.2 ± 0.9 | 20.8 ± 0.6 | 2.4 ± 0.5 | 7.7 ± 0.2 | 15.8 ± 1.0 | 1.9 ± 0.6 | 8.0 ± 0.9 | 13.3 ± 0.8 | 3.9 ± 0.5 | 10.9 ± 0.5 | 15.3 ± 1.3 |
| alphabetically_last_5 | 2.5 ± 0.7 | 6.0 ± 1.3 | 9.9 ± 0.8 | 1.6 ± 0.3 | 5.5 ± 0.4 | 8.5 ± 0.3 | 1.8 ± 0.4 | 6.1 ± 0.8 | 7.8 ± 0.5 | 2.5 ± 0.8 | 6.7 ± 0.6 | 9.3 ± 0.4 |
| capitalize | 8.0 ± 1.0 | 20.7 ± 1.4 | 54.8 ± 2.1 | 3.3 ± 0.3 | 13.2 ± 1.3 | 33.4 ± 1.3 | 3.7 ± 0.1 | 13.5 ± 1.3 | 39.2 ± 2.7 | 6.0 ± 1.6 | 14.7 ± 1.3 | 33.6 ± 2.1 |
| capitalize_first_letter | 10.1 ± 1.2 | 12.5 ± 1.8 | 28.6 ± 1.1 | 5.3 ± 1.0 | 13.4 ± 1.7 | 13.7 ± 1.2 | 5.2 ± 0.3 | 11.2 ± 1.2 | 17.4 ± 0.8 | 8.9 ± 0.7 | 10.0 ± 1.1 | 20.4 ± 1.2 |
| capitalize_last_letter | 4.8 ± 1.3 | 9.6 ± 1.0 | 8.3 ± 1.2 | 8.6 ± 1.1 | 7.5 ± 1.8 | 8.6 ± 0.9 | 9.1 ± 1.4 | 9.3 ± 1.2 | 7.3 ± 1.5 | 5.5 ± 0.7 | 5.8 ± 1.4 | 6.7 ± 0.7 |
| choose_first_of_3 | 10.3 ± 2.3 | 25.5 ± 1.8 | 69.4 ± 1.8 | 6.1 ± 0.9 | 19.0 ± 1.7 | 52.4 ± 2.0 | 4.1 ± 0.7 | 19.1 ± 1.8 | 46.0 ± 2.0 | 11.3 ± 0.7 | 35.2 ± 1.7 | 54.1 ± 1.7 |
| choose_first_of_5 | 8.7 ± 1.3 | 19.7 ± 1.9 | 55.5 ± 1.3 | 4.9 ± 0.7 | 15.3 ± 0.9 | 42.6 ± 3.0 | 3.6 ± 0.7 | 16.2 ± 1.5 | 32.7 ± 1.6 | 7.9 ± 1.0 | 28.2 ± 2.4 | 42.1 ± 1.4 |
| choose_last_of_3 | 2.0 ± 0.4 | 3.2 ± 0.2 | 4.4 ± 1.0 | 1.4 ± 0.4 | 2.8 ± 0.3 | 5.4 ± 0.9 | 1.4 ± 0.5 | 3.0 ± 0.5 | 5.0 ± 0.5 | 1.7 ± 0.3 | 3.0 ± 0.5 | 5.2 ± 0.3 |
| choose_last_of_5 | 1.5 ± 0.4 | 2.9 ± 0.6 | 3.9 ± 1.1 | 1.7 ± 0.5 | 2.4 ± 0.4 | 4.6 ± 0.6 | 1.1 ± 0.4 | 2.9 ± 0.5 | 5.6 ± 0.5 | 1.5 ± 0.3 | 2.6 ± 0.4 | 5.1 ± 0.4 |
| choose_middle_of_3 | 1.7 ± 0.6 | 3.3 ± 0.5 | 4.2 ± 0.5 | 2.1 ± 0.8 | 2.2 ± 0.3 | 6.0 ± 0.1 | 1.3 ± 0.5 | 3.1 ± 0.2 | 5.0 ± 0.3 | 1.7 ± 0.7 | 3.4 ± 0.7 | 4.9 ± 0.7 |
| choose_middle_of_5 | 1.6 ± 0.3 | 3.0 ± 0.6 | 2.7 ± 0.4 | 1.6 ± 0.2 | 2.1 ± 0.3 | 3.1 ± 0.6 | 1.7 ± 0.4 | 2.9 ± 0.7 | 3.9 ± 0.7 | 1.7 ± 0.4 | 2.3 ± 0.6 | 4.7 ± 0.6 |
| lowercase_first_letter | 5.5 ± 0.8 | 8.4 ± 0.9 | 28.5 ± 2.2 | 4.1 ± 0.9 | 6.1 ± 1.2 | 20.1 ± 1.6 | 4.7 ± 0.7 | 6.5 ± 0.6 | 20.6 ± 1.1 | 2.2 ± 0.6 | 8.8 ± 0.7 | 13.3 ± 0.8 |
| lowercase_last_letter | 11.1 ± 0.8 | 7.7 ± 0.7 | 10.5 ± 1.1 | 3.6 ± 0.7 | 8.0 ± 0.7 | 9.6 ± 1.0 | 7.4 ± 1.2 | 8.9 ± 1.0 | 13.3 ± 0.5 | 7.8 ± 1.1 | 10.9 ± 1.5 | 8.9 ± 1.2 |
| next_capital_letter | 4.9 ± 1.1 | 4.2 ± 0.9 | 2.4 ± 0.8 | 3.9 ± 0.3 | 4.2 ± 1.0 | 3.6 ± 0.7 | 4.7 ± 0.8 | 4.1 ± 0.7 | 3.1 ± 0.9 | 4.8 ± 0.9 | 3.5 ± 0.9 | 4.2 ± 1.1 |
| next_item | 3.4 ± 2.2 | 8.6 ± 2.5 | 16.8 ± 2.1 | 2.9 ± 1.9 | 5.7 ± 1.3 | 11.8 ± 1.2 | 1.3 ± 0.9 | 7.8 ± 1.1 | 11.8 ± 2.5 | 6.3 ± 0.8 | 9.0 ± 1.6 | 9.4 ± 1.7 |
| prev_item | 2.5 ± 0.8 | 7.7 ± 1.6 | 15.1 ± 2.3 | 2.0 ± 0.8 | 5.9 ± 0.7 | 10.3 ± 1.0 | 1.4 ± 0.5 | 6.6 ± 2.5 | 11.4 ± 2.5 | 5.7 ± 1.3 | 8.4 ± 1.2 | 7.7 ± 2.2 |
| word_length | 9.9 ± 1.0 | 13.3 ± 0.9 | 12.8 ± 1.1 | 13.4 ± 0.8 | 14.0 ± 1.0 | 13.3 ± 0.8 | 12.0 ± 1.3 | 14.3 ± 0.6 | 13.6 ± 1.6 | 10.8 ± 1.1 | 14.7 ± 1.8 | 12.6 ± 0.7 |
| ICL composite (macro) ↑ | 5.3 ± 0.9 | 9.6 ± 1.1 | 20.0 ± 1.3 | 4.1 ± 0.7 | 7.8 ± 0.9 | 15.2 ± 1.1 | 3.8 ± 0.7 | 8.4 ± 1.0 | 14.7 ± 1.2 | 5.2 ± 0.8 | 10.4 ± 1.1 | 14.9 ± 1.1 |

Table 9: Results across model scales (0.13B, 0.5B, 1B) on THE PILE at iso-FLOPs. Copy snippets use span $L$=20. Evaluation is few-shot: 1-shot for both standard LM benchmarks, and function-style probes. We report per-task accuracy (or EM where standard), averaged over three seeds, and the ICL composite (macro-average across tasks). Higher is better.

| | Baseline | | | Induction | | | Anti-induction | | | Balanced | | |
|---|---|---|---|---|---|---|---|---|---|---|---|---|
| | 0.13B | 0.5B | 1B | 0.13B | 0.5B | 1B | 0.13B | 0.5B | 1B | 0.13B | 0.5B | 1B |
| MMLU | 25.8±0.0 | 25.2±0.3 | 26.8±0.0 | 25.8±0.0 | 24.4±0.3 | 26.4±0.0 | 25.8±0.0 | 24.7±0.3 | 26.8±0.0 | 25.9±0.0 | 24.3±0.2 | 27.3±0.0 |
| Winogrande | 49.8±0.8 | 50.3±0.5 | 49.6±0.4 | 48.9±0.3 | 50.2±0.3 | 50.6±0.4 | 50.9±1.2 | 50.4±0.8 | 49.3±0.9 | 50.8±0.3 | 52.0±0.4 | 51.0±0.0 |
| CommonSenseQA | 20.9±0.9 | 20.6±1.0 | 20.5±0.5 | 20.8±0.8 | 20.6±0.9 | 20.6±0.6 | 20.9±0.8 | 20.6±1.0 | 20.7±0.1 | 21.0±0.8 | 20.7±0.9 | 20.6±0.6 |
| PIQA | 56.8±0.4 | 58.7±0.8 | 59.9±0.4 | 55.4±0.2 | 58.8±0.2 | 58.3±0.3 | 55.9±0.1 | 57.4±0.5 | 59.0±1.1 | 55.4±0.9 | 57.3±0.3 | 58.1±0.2 |
| HellaSwag | 26.4±0.1 | 27.0±0.1 | 27.8±0.1 | 26.2±0.1 | 26.7±0.0 | 27.3±0.1 | 26.3±0.1 | 26.7±0.2 | 27.3±0.1 | 26.2±0.1 | 26.6±0.1 | 27.2±0.2 |
| TriviaQA-Wiki | 0.1±0.0 | 0.1±0.0 | 0.3±0.0 | 0.1±0.0 | 0.1±0.0 | 0.1±0.0 | 0.1±0.0 | 0.1±0.0 | 0.1±0.0 | 0.1±0.0 | 0.1±0.0 | 0.2±0.0 |
| BBH (CoT) | 0.0±0.0 | 0.6±0.0 | 3.4±0.0 | 0.1±0.02 | 2.0±0.0 | 4.3±0.0 | 0.4±0.00 | 0.1±0.0 | 0.6±0.1 | 0.4±0.00 | 1.8±1.1 | 2.8±0.0 |
| OpenBookQA | 14.3±0.5 | 15.6±0.7 | 15.8±0.5 | 14.3±0.2 | 14.5±0.8 | 15.5±0.5 | 13.0±0.0 | 14.7±0.2 | 16.7±0.7 | 13.3±0.1 | 13.9±0.1 | 15.6±1.2 |
| ARC-Challenge | 18.4±0.2 | 19.0±0.4 | 18.1±0.1 | 18.7±0.4 | 18.2±0.1 | 18.3±0.2 | 17.7±0.3 | 17.8±0.2 | 18.4±0.5 | 17.5±0.4 | 17.8±0.4 | 17.7±0.7 |
| GPQA | 24.8±0.6 | 25.0±1.0 | 26.2±2.5 | 25.0±0.8 | 24.4±1.3 | 25.2±0.6 | 25.1±1.1 | 25.9±1.2 | 25.7±1.2 | 25.5±0.3 | 24.2±2.1 | 26.0±2.0 |
| GSM-8K | 1.3±0.1 | 1.9±0.3 | 1.3±0.1 | 1.5±0.6 | 2.2±0.3 | 1.5±0.2 | 1.3±0.5 | 1.8±0.2 | 1.6±0.3 | 1.1±0.1 | 2.0±0.3 | 1.5±0.2 |
| MathQA | 20.7±0.3 | 20.4±0.2 | 20.7±0.3 | 20.3±0.5 | 20.2±0.3 | 21.1±0.3 | 20.1±0.1 | 20.3±0.1 | 20.8±0.7 | 20.6±0.4 | 20.2±0.3 | 21.0±1.0 |
| BoolQ | 53.2±0.8 | 53.4±0.8 | 54.2±0.3 | 53.2±0.8 | 57.6±1.1 | 56.0±0.2 | 50.8±0.8 | 53.0±0.7 | 54.1±0.1 | 53.1±0.8 | 53.1±0.7 | 55.2±0.4 |
| LAMBADA | 9.5±0.1 | 11.9±0.1 | 15.1±0.1 | 6.5±0.1 | 9.9±0.2 | 13.4±0.2 | 5.9±0.2 | 9.6±0.3 | 13.5±0.2 | 7.0±0.1 | 9.2±0.1 | 13.8±0.5 |
| ICL composite (macro) | 23.0±0.3 | 23.6±0.4 | 24.3±0.4 | 22.6±0.3 | 23.6±0.4 | 24.2±0.3 | 22.4±0.4 | 23.1±0.4 | 23.9±0.4 | 22.7±0.3 | 23.1±0.5 | 24.1±0.5 |
| alphabetically_first_3 | 2.6±0.6 | 3.0±0.7 | 13.2±1.3 | 3.3±0.7 | 4.8±0.6 | 8.9±0.8 | 1.9±0.4 | 3.5±0.8 | 5.8±1.0 | 3.6±0.7 | 4.4±0.4 | 8.5±0.8 |
| alphabetically_first_5 | 2.7±0.4 | 3.3±0.5 | 8.9±0.5 | 3.3±0.1 | 4.3±0.2 | 7.5±0.4 | 2.0±0.3 | 3.7±0.8 | 4.4±0.8 | 2.2±0.4 | 3.1±0.6 | 6.8±1.1 |
| alphabetically_last_3 | 2.1±0.5 | 2.9±0.5 | 13.3±1.2 | 3.1±0.4 | 4.3±0.9 | 9.2±1.0 | 1.9±0.4 | 2.9±0.3 | 6.8±0.6 | 4.5±0.3 | 3.8±0.5 | 10.6±0.9 |
| alphabetically_last_5 | 1.6±0.5 | 2.8±0.4 | 8.5±0.9 | 1.9±0.6 | 3.2±0.4 | 6.0±0.3 | 1.4±0.4 | 2.3±0.5 | 4.5±0.4 | 2.2±0.4 | 3.0±0.8 | 6.8±0.6 |
| capitalize | 2.9±0.6 | 1.5±0.4 | 4.4±0.7 | 1.8±0.3 | 5.1±0.8 | 2.6±0.8 | 1.8±0.2 | 2.5±0.5 | 4.9±0.6 | 2.7±0.4 | 3.1±0.5 | 3.7±1.1 |
| capitalize_first_letter | 4.1±1.1 | 3.0±0.9 | 3.4±0.7 | 3.5±1.0 | 5.6±1.4 | 4.0±1.2 | 3.5±1.0 | 4.7±1.2 | 4.0±0.9 | 4.4±1.0 | 4.9±0.8 | 3.5±1.1 |
| capitalize_last_letter | 9.0±0.7 | 7.4±0.4 | 5.6±0.9 | 9.1±0.8 | 6.0±0.7 | 8.4±0.4 | 9.2±0.8 | 8.7±0.9 | 8.4±0.8 | 8.7±0.7 | 8.8±0.8 | 8.8±0.5 |
| choose_first_of_3 | 5.6±0.5 | 6.7±1.0 | 37.0±1.2 | 5.5±0.8 | 7.4±1.3 | 25.1±2.1 | 1.5±0.4 | 6.0±1.1 | 14.6±1.0 | 8.9±1.5 | 7.2±0.8 | 26.1±1.8 |
| choose_first_of_5 | 5.9±0.4 | 6.9±0.8 | 32.0±1.2 | 3.9±0.8 | 5.6±0.6 | 24.9±1.9 | 1.1±0.3 | 6.1±0.8 | 12.5±1.9 | 5.3±0.3 | 4.8±0.6 | 23.8±1.4 |
| choose_last_of_3 | 0.8±0.3 | 1.1±0.3 | 3.7±0.6 | 1.3±0.4 | 1.9±0.3 | 2.4±0.5 | 1.2±0.4 | 1.3±0.7 | 2.0±0.4 | 0.7±0.2 | 1.9±0.7 | 3.7±0.3 |
| choose_last_of_5 | 0.9±0.3 | 1.1±0.4 | 2.4±0.4 | 1.4±0.3 | 1.5±0.3 | 2.0±0.4 | 0.9±0.3 | 1.3±0.4 | 1.9±0.6 | 0.6±0.2 | 1.3±0.4 | 3.1±0.6 |
| choose_middle_of_3 | 0.7±0.3 | 1.1±0.4 | 3.5±0.6 | 0.9±0.1 | 1.6±0.4 | 2.5±0.6 | 0.7±0.2 | 1.1±0.1 | 1.9±0.5 | 0.6±0.1 | 1.1±0.2 | 3.6±0.4 |
| choose_middle_of_5 | 0.9±0.4 | 1.4±0.5 | 2.3±0.5 | 1.2±0.3 | 1.4±0.2 | 2.2±0.5 | 1.2±0.3 | 1.3±0.5 | 1.6±0.4 | 0.8±0.4 | 1.5±0.7 | 3.0±0.6 |
| lowercase_first_letter | 4.9±0.7 | 3.7±0.7 | 4.0±0.3 | 4.0±0.7 | 3.9±0.6 | 4.6±0.8 | 2.6±0.5 | 4.7±0.7 | 4.7±0.8 | 2.3±0.5 | 4.7±0.7 | 4.4±0.8 |
| lowercase_last_letter | 10.6±1.2 | 9.3±1.2 | 9.5±1.2 | 6.6±0.9 | 7.7±1.3 | 10.6±1.2 | 7.6±0.5 | 10.6±1.2 | 10.6±1.2 | 8.7±1.3 | 10.6±1.2 | 10.3±1.2 |
| next_capital_letter | 4.5±0.3 | 3.8±0.6 | 3.6±1.0 | 4.6±0.5 | 3.9±0.4 | 4.5±0.4 | 4.4±0.7 | 4.7±0.1 | 4.1±0.5 | 4.1±0.4 | 4.5±0.4 | 4.6±0.5 |
| next_item | 2.5±0.4 | 2.0±1.7 | 9.7±1.0 | 3.9±0.8 | 5.1±1.4 | 5.1±1.2 | 1.5±1.1 | 2.8±1.1 | 3.5±0.8 | 4.4±2.1 | 4.2±1.1 | 3.4±1.5 |
| prev_item | 2.5±1.2 | 2.3±1.7 | 9.2±1.6 | 3.0±1.1 | 5.1±1.6 | 5.7±2.0 | 1.8±0.7 | 3.3±1.9 | 3.4±1.9 | 4.6±0.7 | 3.0±1.5 | 3.8±1.3 |
| word_length | 13.8±1.9 | 13.7±1.7 | 12.7±2.2 | 12.7±1.5 | 13.4±2.1 | 13.6±1.5 | 12.5±1.6 | 13.9±1.7 | 13.9±1.4 | 10.6±1.0 | 13.9±1.7 | 13.9±1.9 |
| ICL composite (macro) ↑ | 4.1±0.6 | 4.1±0.8 | 9.8±0.9 | 3.9±0.6 | 4.8±0.8 | 7.9±1.0 | 3.1±0.5 | 4.5±0.8 | 6.0±0.9 | 4.2±0.6 | 4.7±0.7 | 8.0±1.0 |

Table 10: Function-probe suite of Todd et al. (2024) under label-permutation stress (Todd et al. (2024) suite): HITS@1 accuracy on 10-shot prompts with demonstration labels randomly permuted; reported as mean±std across three seeds for 0.13B, 0.5B, and 1B, comparing Baseline, Induction, Anti, and Balanced curricula.

| | Baseline | | | Induction | | | Anti-induction | | | Balanced | | |
|---|---|---|---|---|---|---|---|---|---|---|---|---|
| | 0.13B | 0.5B | 1B | 0.13B | 0.5B | 1B | 0.13B | 0.5B | 1B | 0.13B | 0.5B | 1B |
| alphabetically_first_3 | 4.4±0.4 | 9.2±0.7 | 16.6±0.6 | 4.4±0.3 | 7.0±0.7 | 12.3±1.4 | 3.6±0.5 | 8.2±0.8 | 11.0±1.3 | 4.2±0.7 | 10.7±1.4 | 12.5±0.8 |
| alphabetically_first_5 | 4.1±0.7 | 7.1±1.1 | 11.1±0.9 | 4.2±0.6 | 5.9±0.4 | 9.4±0.5 | 2.6±0.9 | 7.0±0.9 | 7.0±0.4 | 3.5±0.4 | 7.8±1.0 | 9.0±0.5 |
| alphabetically_last_3 | 3.2±0.6 | 9.7±0.8 | 16.1±1.6 | 2.6±0.6 | 7.3±0.9 | 12.3±0.6 | 2.7±1.0 | 7.4±1.3 | 10.4±0.9 | 4.8±0.6 | 10.7±1.4 | 13.0±0.6 |
| alphabetically_last_5 | 2.6±0.3 | 5.8±0.9 | 9.5±0.2 | 1.6±0.3 | 5.3±0.3 | 7.8±0.7 | 1.6±0.5 | 5.5±0.6 | 6.4±0.8 | 2.0±0.2 | 6.7±0.5 | 8.5±0.9 |
| capitalize | 8.7±0.8 | 18.2±1.9 | 49.3±2.9 | 3.6±1.0 | 13.3±1.8 | 29.9±0.8 | 4.3±0.7 | 13.5±1.6 | 34.3±1.4 | 6.8±0.6 | 14.5±0.9 | 31.0±1.9 |
| capitalize_first_letter | 10.4±2.1 | 13.2±1.4 | 29.0±1.4 | 6.4±0.4 | 13.3±0.8 | 15.2±1.4 | 6.4±0.9 | 12.2±1.9 | 17.5±1.3 | 9.2±1.5 | 10.8±2.2 | 21.2±0.7 |
| capitalize_last_letter | 4.7±1.0 | 8.2±1.0 | 7.7±0.7 | 7.9±1.2 | 6.7±0.8 | 7.4±1.3 | 8.9±1.7 | 8.6±0.9 | 5.9±1.4 | 5.2±0.9 | 5.3±0.9 | 6.5±1.4 |
| choose_first_of_3 | 11.2±1.2 | 21.9±2.1 | 54.9±2.6 | 6.9±1.2 | 19.1±2.5 | 35.0±2.5 | 5.2±0.4 | 19.0±2.0 | 28.9±0.8 | 13.4±1.0 | 32.4±1.1 | 34.5±1.6 |
| choose_first_of_5 | 9.3±1.5 | 17.3±1.6 | 42.0±2.3 | 5.6±0.9 | 15.3±1.8 | 28.9±2.1 | 3.6±0.4 | 16.8±1.9 | 20.3±1.3 | 8.5±1.7 | 23.7±1.9 | 26.4±1.9 |
| choose_last_of_3 | 1.6±0.4 | 2.7±0.6 | 4.3±0.9 | 2.1±0.6 | 2.6±0.4 | 4.9±0.5 | 1.3±0.4 | 2.8±0.3 | 4.9±1.2 | 1.3±0.3 | 3.3±0.7 | 5.4±0.6 |
| choose_last_of_5 | 1.7±0.4 | 2.2±0.6 | 3.9±0.8 | 1.9±0.4 | 2.1±0.3 | 4.5±0.6 | 1.1±0.4 | 2.3±0.6 | 5.5±0.7 | 1.2±0.5 | 2.3±0.8 | 4.7±0.3 |
| choose_middle_of_3 | 1.9±0.3 | 3.5±0.5 | 4.5±0.9 | 2.3±0.6 | 2.8±0.7 | 5.6±1.1 | 1.4±0.5 | 3.0±0.4 | 4.6±0.3 | 1.5±0.3 | 3.1±0.3 | 5.2±0.8 |
| choose_middle_of_5 | 1.8±0.7 | 3.6±0.8 | 2.9±0.5 | 1.9±0.3 | 2.2±0.6 | 3.3±0.4 | 1.5±0.6 | 2.8±0.4 | 3.9±0.5 | 1.7±0.2 | 2.4±0.4 | 4.6±0.7 |
| lowercase_first_letter | 6.7±0.8 | 9.3±0.6 | 27.8±1.6 | 4.9±0.7 | 6.5±0.4 | 19.0±1.1 | 5.4±1.1 | 7.5±1.1 | 18.4±1.9 | 2.5±0.6 | 10.4±0.9 | 13.9±1.4 |
| lowercase_last_letter | 10.1±1.0 | 7.5±1.2 | 8.8±1.9 | 3.5±1.2 | 8.0±1.6 | 8.2±1.2 | 7.1±0.9 | 8.4±1.7 | 12.8±1.8 | 7.1±0.7 | 9.5±1.3 | 7.8±1.5 |
| next_capital_letter | 4.3±0.9 | 4.3±0.5 | 2.8±0.4 | 4.2±0.6 | 3.5±0.6 | 3.6±0.6 | 4.7±0.6 | 4.2±0.7 | 3.0±0.6 | 4.3±0.5 | 3.3±0.6 | 3.0±0.1 |
| next_item | 3.3±1.0 | 7.3±1.3 | 15.3±1.8 | 2.7±0.8 | 4.7±0.8 | 10.8±2.2 | 1.8±1.0 | 4.9±1.2 | 9.7±2.7 | 5.3±1.5 | 6.6±1.4 | 7.2±1.0 |
| prev_item | 3.2±1.7 | 6.5±2.3 | 13.5±1.6 | 2.5±1.4 | 5.6±1.4 | 8.6±1.8 | 2.3±1.5 | 4.9±1.1 | 9.2±1.1 | 4.8±1.7 | 6.5±1.4 | 7.8±1.0 |
| word_length | 9.7±0.8 | 12.6±1.3 | 11.4±1.8 | 12.8±1.0 | 12.9±1.4 | 12.4±1.1 | 12.1±1.1 | 14.5±1.6 | 13.2±0.2 | 11.1±0.9 | 14.3±0.8 | 12.6±1.0 |
| ICL composite (macro) ↑ | 5.4±0.9 | 9.0±1.1 | 17.4±1.3 | 4.3±0.7 | 7.6±1.0 | 12.6±1.2 | 4.1±0.8 | 8.1±1.1 | 11.9±1.1 | 5.2±0.8 | 9.7±1.0 | 12.4±1.0 |

Table 11: Function-probe suite of Todd et al. (2024) under decision-rule sensitivity: HITS@3 accuracy on 10-shot prompts, reported as mean±std across three seeds, for 0.13B, 0.5B, and 1B, comparing Baseline, Induction, Anti, and Balanced curricula.

| | Baseline | | | Induction | | | Anti-induction | | | Balanced | | |
|---|---|---|---|---|---|---|---|---|---|---|---|---|
| | 0.13B | 0.5B | 1B | 0.13B | 0.5B | 1B | 0.13B | 0.5B | 1B | 0.13B | 0.5B | 1B |
| alphabetically_first_3 | 8.8 ± 0.8 | 15.3 ± 0.5 | 31.0 ± 0.5 | 8.6 ± 1.3 | 12.7 ± 0.3 | 23.0 ± 0.8 | 7.1 ± 0.3 | 13.7 ± 0.8 | 22.6 ± 1.4 | 9.3 ± 1.2 | 18.6 ± 0.6 | 24.1 ± 0.8 |
| alphabetically_first_5 | 8.6 ± 1.3 | 13.0 ± 0.5 | 20.7 ± 1.5 | 8.3 ± 0.9 | 12.1 ± 1.0 | 16.4 ± 0.6 | 7.4 ± 0.7 | 13.6 ± 1.3 | 15.5 ± 1.3 | 7.8 ± 1.3 | 13.3 ± 0.6 | 15.9 ± 0.8 |
| alphabetically_last_3 | 7.3 ± 0.9 | 17.4 ± 0.7 | 30.3 ± 1.0 | 6.3 ± 0.7 | 14.3 ± 0.5 | 24.5 ± 1.1 | 5.7 ± 0.7 | 14.6 ± 1.5 | 22.3 ± 1.5 | 8.9 ± 0.3 | 19.1 ± 1.1 | 24.0 ± 0.9 |
| alphabetically_last_5 | 5.8 ± 0.8 | 12.6 ± 1.7 | 18.6 ± 1.1 | 4.9 ± 0.8 | 11.4 ± 0.8 | 14.6 ± 0.7 | 4.5 ± 0.5 | 10.9 ± 1.4 | 14.0 ± 1.0 | 5.6 ± 0.8 | 12.9 ± 0.9 | 15.6 ± 1.0 |
| capitalize | 17.8 ± 1.1 | 39.7 ± 1.6 | 73.8 ± 1.4 | 10.6 ± 0.4 | 28.5 ± 0.9 | 57.7 ± 1.3 | 11.5 ± 1.2 | 28.9 ± 1.3 | 60.6 ± 0.8 | 14.0 ± 2.0 | 32.2 ± 1.4 | 56.0 ± 0.8 |
| capitalize_first_letter | 25.6 ± 1.6 | 34.7 ± 1.8 | 55.2 ± 2.0 | 17.1 ± 1.1 | 30.7 ± 1.7 | 40.1 ± 1.6 | 19.1 ± 1.5 | 29.9 ± 1.7 | 43.5 ± 2.1 | 22.3 ± 1.7 | 29.6 ± 1.3 | 42.0 ± 2.1 |
| capitalize_last_letter | 14.1 ± 1.7 | 25.6 ± 1.7 | 20.4 ± 2.8 | 20.0 ± 1.1 | 18.8 ± 1.9 | 23.0 ± 2.4 | 20.8 ± 2.1 | 23.2 ± 2.8 | 17.2 ± 1.8 | 15.6 ± 0.8 | 15.9 ± 1.7 | 18.6 ± 2.7 |
| choose_first_of_3 | 18.3 ± 1.9 | 38.6 ± 1.9 | 83.6 ± 0.6 | 12.9 ± 1.0 | 27.8 ± 1.7 | 66.6 ± 1.6 | 9.8 ± 1.1 | 28.3 ± 1.2 | 60.7 ± 0.9 | 19.5 ± 1.1 | 50.3 ± 2.1 | 67.7 ± 1.0 |
| choose_first_of_5 | 17.1 ± 1.7 | 30.6 ± 1.9 | 74.1 ± 1.1 | 11.3 ± 0.7 | 24.4 ± 1.6 | 56.2 ± 1.3 | 8.0 ± 1.3 | 26.3 ± 1.7 | 48.3 ± 1.5 | 15.4 ± 1.2 | 41.5 ± 1.8 | 56.7 ± 1.8 |
| choose_last_of_3 | 5.1 ± 0.4 | 8.0 ± 0.6 | 11.0 ± 1.0 | 4.5 ± 0.9 | 8.7 ± 0.7 | 11.5 ± 1.3 | 3.9 ± 0.5 | 7.9 ± 0.6 | 11.6 ± 0.6 | 4.0 ± 0.6 | 8.7 ± 1.2 | 12.4 ± 0.4 |
| choose_last_of_5 | 4.0 ± 0.6 | 7.1 ± 0.7 | 9.3 ± 1.1 | 4.9 ± 0.6 | 7.0 ± 0.8 | 10.3 ± 0.7 | 3.5 ± 0.6 | 7.6 ± 0.8 | 10.1 ± 0.6 | 4.1 ± 0.7 | 6.6 ± 0.4 | 10.6 ± 1.0 |
| choose_middle_of_3 | 4.8 ± 1.1 | 8.7 ± 0.7 | 12.5 ± 1.0 | 5.5 ± 0.4 | 7.7 ± 0.7 | 13.1 ± 0.7 | 4.0 ± 0.2 | 7.8 ± 0.4 | 10.3 ± 0.7 | 4.4 ± 0.6 | 8.1 ± 1.1 | 11.4 ± 0.7 |
| choose_middle_of_5 | 4.6 ± 0.7 | 7.6 ± 0.5 | 7.7 ± 0.8 | 4.5 ± 0.2 | 6.2 ± 0.8 | 7.4 ± 0.9 | 4.5 ± 0.9 | 7.4 ± 1.1 | 8.4 ± 0.7 | 4.2 ± 0.8 | 6.5 ± 0.9 | 9.3 ± 1.0 |
| lowercase_first_letter | 19.4 ± 2.0 | 28.7 ± 0.9 | 58.6 ± 2.3 | 11.4 ± 1.0 | 22.1 ± 1.4 | 49.5 ± 1.9 | 12.4 ± 1.7 | 22.8 ± 3.2 | 45.1 ± 2.2 | 5.5 ± 0.3 | 30.5 ± 1.6 | 38.7 ± 0.5 |
| lowercase_last_letter | 28.0 ± 1.4 | 19.4 ± 1.1 | 23.5 ± 1.2 | 8.7 ± 1.2 | 23.1 ± 0.6 | 26.2 ± 1.3 | 20.2 ± 1.1 | 24.0 ± 0.9 | 29.7 ± 0.9 | 13.2 ± 1.2 | 26.8 ± 2.0 | 22.8 ± 1.8 |
| next_capital_letter | 15.0 ± 0.5 | 13.2 ± 1.2 | 12.5 ± 0.7 | 13.2 ± 0.9 | 12.4 ± 1.8 | 11.7 ± 1.4 | 16.0 ± 1.0 | 13.7 ± 1.4 | 10.9 ± 2.0 | 12.5 ± 0.8 | 11.1 ± 1.4 | 13.3 ± 1.5 |
| next_item | 8.5 ± 2.2 | 18.7 ± 3.9 | 29.5 ± 3.2 | 9.0 ± 1.9 | 15.8 ± 1.9 | 23.4 ± 0.8 | 5.8 ± 2.0 | 18.2 ± 3.4 | 25.4 ± 1.9 | 13.5 ± 1.2 | 20.5 ± 3.0 | 20.3 ± 2.1 |
| prev_item | 8.4 ± 0.8 | 15.1 ± 1.4 | 24.7 ± 2.4 | 7.6 ± 1.2 | 14.9 ± 2.1 | 17.3 ± 1.3 | 5.3 ± 1.8 | 17.0 ± 2.9 | 20.1 ± 3.7 | 13.8 ± 0.8 | 17.1 ± 2.1 | 17.6 ± 0.8 |
| word_length | 31.0 ± 2.2 | 36.5 ± 2.1 | 37.6 ± 0.6 | 33.4 ± 1.4 | 40.2 ± 2.2 | 38.4 ± 2.2 | 28.5 ± 1.1 | 39.9 ± 1.9 | 40.0 ± 3.2 | 31.5 ± 1.0 | 41.1 ± 2.4 | 36.6 ± 1.1 |
| ICL composite (macro) ↑ | 13.3 ± 1.2 | 20.6 ± 1.3 | 33.4 ± 1.4 | 10.7 ± 0.9 | 17.8 ± 1.2 | 27.9 ± 1.3 | 10.4 ± 1.1 | 18.7 ± 1.6 | 27.2 ± 1.5 | 11.8 ± 1.0 | 21.6 ± 1.5 | 27.0 ± 1.2 |

Table 12: Function-probe suite of Todd et al. (2024): HITS@1 accuracy on 10-shots prompts, reported as mean±std across three seeds, for 0.13B, 0.5B, and 1B, comparing Clean run, top-2% induction heads drop (↓**Induct Hd 2%**), random heads drop (↓**Rand**) across Baseline, Induction, Anti, and Balanced curricula.

