# OpenReview forum: "What Matters More For In-Context Learning under Matched Compute Budgets: Pretraining on Natural Text or Incorporating Targeted Synthetic Examples?"
_ICLR.cc/2026/Conference — ICLR 2026 Conference Withdrawn Submission_

### Official Review · Reviewer_L7QA · 2025-10-24

**Soundness:** 2
**Presentation:** 2
**Contribution:** 1
**Rating:** 2
**Confidence:** 4

**Summary:**

This paper proposes Bi-Induct, which attempts to accelerate the emergence of the induction circuit during training through adjustments to the training data. The study finds that models trained with Bi-Induct (1) do not necessarily exhibit better ICL performance, and (2) tend to develop more redundant induction circuits.

**Strengths:**

1. The research question posed by the authors, i.e., how to accelerate the emergence of a known circuit through improvements in training methodology, is meaningful. Achieving this goal would significantly advance the field of mechanistic interpretability. The paper’s attempt to explicitly link data modifications with mechanistic emergence is interesting, and such exploration could be valuable if supported by stronger empirical evidence.

2. The paper’s discussion of induction circuit redundancy, i.e., the extent to which ablating top induction heads leads to performance degradation, is interesting. Moreover, their experimental approach of ablating the top 2% of induction heads is reasonable. Measuring the redundancy of various known circuits and examining how this redundancy changes with model depth or width is an intriguing direction.

**Weaknesses:**

1. The main conclusion of this paper: “Natural data yields load-bearing heads, Bi-Induct induces more distributed, redundant heads.” has limited impact. The fact that natural data induces induction heads has already been established by Olsson et al., and the significance of Bi-Induct is limited. Specifically, the paper does not clearly define the ultimate objective of the proposed Bi-Induct training framework. If it is intended as an experimental probing setup, then what kind of real-world training or inference dynamics does it simulate or reveal? If, instead, it is meant as an applied setting for enhancing ICL, the experimental results do not demonstrate any clear empirical advantage. Consequently, I cannot fully grasp the importance of the paper’s core conclusion, it seems to be an analysis conducted under an artificial setup that is not particularly distinctive.

2. The paper claims that “Accelerating the onset of this transition could reduce compute and expose internal circuits for analysis earlier.” and therefore proposes the Bi-Induct training method. However, I could not find any discussion of training dynamics in the main text that demonstrates Bi-Induct indeed triggers the induction circuit earlier. A simple plot of **induction scores against training steps** could easily verify this. In Line 320, the authors discuss the layer positions of induction heads within inference dynamics, but this is not equivalent to “accelerating the onset of this transition”, since in Olsson et al.’s work, the phase transition is defined with respect to training steps. Hence, the “Key empirical finding: Early induction ≠ better ICL.” stated in Line 084 also lacks sufficient support.

3. Some experimental settings lack justification.

    - If the goal of the experiments is to accelerate induction formation for higher training efficiency or to contrast with slow induction emergence, then improving the training objective (for example, introducing a term that encourages prefix-matching patterns in certain attention heads in the loss function) would be more intuitive and practical than modifying the training data stream. Please explain what advantages data stream modification provides over the training objective modification.

    - I do not understand the design purpose of Back-Induction. Compared with typical induction (prefix matching), the input format of Back-Induction seems unrelated to the format of ICL inputs. Therefore, I am unclear about the motivation for including this experiment. Was it merely a form of controlled variable analysis?

    - The linear annealing schedule for synthetic sequence mixing is also confusing. The authors claim some benefits of such linear annealing in Line 207, but it is not very convincing. E.g., “concentrating copy cues early helps trigger the phase transition without interfering with late-stage calibration”, where you can utilize a constant $m(t)$ in the early steps, and turn it off in later steps.

    - The authors conduct experiments on three model scales, but three points are too few to robustly observe scaling trends against model size. I suggest repeating the experiments on more model scales. Furthermore, I suggest a stronger variable control in model configuration, such as fixing width while increasing depth, or fixing depth while increasing width.

**Questions:**

See weakness.

---

### Official Review · Reviewer_1hRE · 2025-10-28

**Soundness:** 3
**Presentation:** 3
**Contribution:** 3
**Rating:** 4
**Confidence:** 4

**Summary:**

This paper presents a carefully conducted, mechanistic investigation into a timely and important question: whether targeted data interventions can effectively shape the emergence of fundamental circuits like induction heads, and if this leads to better in-context learning (ICL) performance under a fixed compute budget.

**Strengths:**

1.  Significant. The results challenge a simple "earlier-is-better" or "synthetic-is-better" intuition. The distinction between circuit emergence and circuit load-bearing is a valuable conceptual contribution to the field. The head ablation results (Table 3) are particularly convincing in demonstrating this point.

2. Presentation is well. The paper is well-written, the figures are clear, and the appendix is thorough, providing necessary details for reproducibility (e.g., span length and mix ratio sweeps).

**Weaknesses:**

1. Generalizability of Findings: The study is necessarily focused on the induction head circuit. While this is a canonical and important circuit for ICL, such as function vector [1].  Naive induction or anti introduction in the vocabulary space may seem naive, but I think it would be more meaningful if induction could be done in an abstract space.

2. **Limitations of evaluation.** According to Table 6 in the Appendix, we can find that the scores of MMLU and ARC-C are actually **random**. It is also difficult to expect a model of this scale to achieve good results, so the final outcome may be influenced by these random benchmarks.



[1] https://arxiv.org/pdf/2310.15213

**Questions:**

1.The most important thing is the evaluation. Can you eliminate almost random or zero point benchmarks?

2.It can be noted that the experiments were conducted on the causal mask model. So, what potential impact does the causal mask have on the experimental conclusions?

3.The discussion mentions a "pathway shift" in larger models towards FFN/residual pathways. Do you have any evidence from your experiments (e.g., probing FFN layers) to support this hypothesis, or is it primarily informed by prior work? why might natural text be more effective at creating "load-bearing" circuits? Is it the compositional structure, the correlation of copy patterns with meaningful linguistic constructs, or the need for the model to use tokens in multiple, competing ways that force efficient circuit design? A brief discussion of these potential factors would strengthen the paper.

---

### Official Review · Reviewer_7P1L · 2025-10-28

**Soundness:** 3
**Presentation:** 2
**Contribution:** 2
**Rating:** 2
**Confidence:** 3

**Summary:**

The paper studies an important question of whether the pretrained natural text or synthetic examples help more with ICL ability. The paper introduces the experiment in which a set of well constructed samples of two major types (and their mixture), namely induction and anti-induction, are introduced along with the normal data, whose downstream ICL performance is probed. The results show that although for 1B scale the quality is comparable (while smaller models benefit more from natural data baseline), the important finding is that Bi-Induct data makes the attention heads more distributed redundant heads.

**Strengths:**

1. The paper proposes a principled way of studying the effect of injecting synthetic induction and anti-induction data for ICL.
2. It includes models of different sizes from 0.13B to 1B.
3. Both perplexity and few-shot downstream tasks are included.
4. A very well written studies that try to explain some of the observations

**Weaknesses:**

1. Most of the ICL ability emerges when the model size becomes large enough. However, the focus of this work stops at 1B, which the review believes has not yet reached the point of where ICL really shines.
2. The copy-style snippets injected into the pretraining stream seem too constrained and simplified, and the review still feels skeptical about its practicality on larger scale experiments.
3. The paper studies an interesting problem itself but the results are well within expectation (with no improvements on quality etc despite the fact that the authors already state that this is more of a principled study) and the main key finding here does not directly lead to any clear specific future techniques. It would be of a much stronger paper if the experiment can be simplified and use the main finding on attention heads as a motivation to derive a potential technique for improving the model performance.
4. The writing style makes it really hard to understand the relatively simple setup and results. The reviewer feels like it would be better to simplify the narrative and focus on a few key findings as well as their implications.
5. Overall, while the study itself is extremely important and interesting, the reviewer does not feel convinced that its potential influence is well stated in the sense that it can benefit from future research.

**Questions:**

Listed above in weakness.

---

### Official Review · Reviewer_khsG · 2025-10-28

**Soundness:** 3
**Presentation:** 2
**Contribution:** 2
**Rating:** 4
**Confidence:** 2

**Summary:**

The authors introduce “bi-induct” - a strategy for inserting pairs of token sequences separated by a whitespace into the pre-training data stream. They show that it can manipulate the occurrence of induction heads in a language model - attention heads that are believed to be responsible for copying part of the context into the model output.

Using this mechanism, they study the hypothesis that the appearance of induction heads early (shallower) in the model directly translates to better ICL. This hypothesis is based on the intuition that some researchers in the community might hold.

In a series of experiments, they disproved this hypothesis.

**Strengths:**

The authors have presented a counterargument to an intuition of some researchers in the mechanistic interpretability community, which could impact the focus of further research on ICL..

They suggest a methodology for manipulating the appearance of induction heads. Their methodology could be reused in further studies of phenomena related to induction heads.

**Weaknesses:**

Although the experiments are well-documented, the lack of open code to reproduce the experiments reduces the confidence in the presented empirical results.

The overall structure of the paper made it difficult to read. A lot of the material is lumped together without a clear distinction between important parts and secondary analysis.

**Questions:**

If the 1B natural-only baseline exhibits earlier and broader consolidation of induction, does that imply that the target synthetic strategy is not scalable?

Given that natural data concentrates the induction responsibility on subset of heads, while Bi-Induct distributes it more broadly, the ICL downstream performance does not significantly improve. Is this because the highly active head lose their ability to perform ICL, or because other heads lose their capability due to multi-taking?

---

### Note · Authors · 2025-11-20

I have read and agree with the venue's withdrawal policy on behalf of myself and my co-authors.